# 16p11.2 deletion is associated with hyperactivation of human iPSC-derived dopaminergic neuron networks and is rescued by RHOA inhibition in vitro

Maria Sundberg [1], Hannah Pinson [2,3,11], Richard S. Smith [4,11], Kellen D. Winden [1], Pooja Venugopal [1], Derek J. C. Tai [5,6], James F. Gusella [5,7,8,9], Michael E. Talkowski [5,9,10], Christopher A. Walsh [4], Max Tegmark [2] & Mustafa Sahin [1✉]

Reciprocal copy number variations (CNVs) of 16p11.2 are associated with a wide spectrum of neuropsychiatric and neurodevelopmental disorders. Here, we use human induced pluripotent stem cells (iPSCs)-derived dopaminergic (DA) neurons carrying CNVs of 16p11.2 duplication (16pdup) and 16p11.2 deletion (16pdel), engineered using CRISPR-Cas9. We show that 16pdel iPSC-derived DA neurons have increased soma size and synaptic marker expression compared to isogenic control lines, while 16pdup iPSC-derived DA neurons show deficits in neuronal differentiation and reduced synaptic marker expression. The 16pdel iPSC-derived DA neurons have impaired neurophysiological properties. The 16pdel iPSC-derived DA neuronal networks are hyperactive and have increased bursting in culture compared to controls. We also show that the expression of RHOA is increased in the 16pdel iPSC-derived DA neurons and that treatment with a specific RHOA-inhibitor, Rhosin, rescues the network activity of the 16pdel iPSC-derived DA neurons. Our data suggest that 16p11.2 deletion-associated iPSC-derived DA neuron hyperactivation can be rescued by RHOA inhibition.

[1] Department of Neurology, F.M. Kirby Neurobiology Center, Rosamund Stone Zander Translational Neuroscience Center, Boston Children's Hospital, Harvard Medical School, Boston, MA, USA. [2] Department of Physics and Center for Brains, Minds & Machines, MIT, Cambridge, MA, USA. [3] Applied Physics/Data Analytics, Vrije Universiteit Brussel, Brussels, Belgium. [4] Division of Genetics and Genomics, Manton Center for Orphan Disease Research, and Howard Hughes Medical Institute, Boston Children's Hospital, Harvard Medical School, Boston, MA, USA. [5] Center for Genomic Medicine, Massachusetts General Hospital, Boston, MA, USA. [6] Department of Neurology, Massachusetts General Hospital, Harvard Medical School, Boston, MA, USA. [7] Department of Genetics, Blavatnik Institute, Harvard Medical School, Boston, MA, USA. [8] Harvard Stem Cell Institute, Harvard University, Cambridge, MA, USA. [9] Program in Medical and Population Genetics Broad Institute of MIT and Harvard, Cambridge, MA, USA. [10] Stanley Center for Psychiatric Research, Broad Institute of MIT and Harvard, Cambridge, MA, USA. [11] These authors contributed equally: Pinson, H., Smith, R. ✉email: mustafa.sahin@childrens.harvard.edu

Autism and schizophrenia (SCZ) are neuropsychiatric disorders manifesting with increasing numbers worldwide. To develop effective treatments for these disorders, detailed genetic and molecular information concerning patients' brain development and network function are needed. Several genome-wide studies of large patient cohorts have shown that recurrent, reciprocal copy number variations (CNVs) of several genomic segments are strongly associated with autism and SCZ risk. Among these segments, reciprocal CNV of 16p11.2 plays a key role in the occurrence of these disorders[1,2]. A meta-analysis of 3613 patients with idiopathic autism spectrum disorder (ASD) showed that 0.50% of patients had 16p11.2 deletion (16pdel) and 0.28% had 16p11.2 duplication (16pdup)[1], as compared with a 0.04–0.05% in the control populations[3,4]. Overall, patients with 16pdel have an increased risk of developmental delay and ASD[5], and behavioral problems associated with this CNV also include attention deficit hyperactivity disorder[6]. Another meta-analysis of 6882 SCZ patients and 6316 controls determined that 0.35% of the patients had 16pdup, as compared with a 0.03% in the control population[2]. In addition to SCZ, patients with 16pdup often have an increased risk of bipolar disorder, anxiety, and depression[7–9].

In CNV disorders, cellular deficits may arise from synergistic effects of the genes located in the CNV region. The 16p11.2 region contains 29 coding genes, and the majority of which are expressed in the brain and involved in neuronal development. However, some of the pathways activated by the 16p11.2 locus may have more prominent roles in brain development than the others. As an example, TAOK2 (TAO kinase 2) regulates synaptic expression, axonal and dendritic formation, and spine development in neurons[10]. MAPK3 (mitogen-activated protein kinase 3) regulates the Ras/MAPK pathway, and it is associated with ERK (extracellular signal-regulated kinase) pathway activation that affects neural differentiation and proliferation[11]. KCTD13 (potassium channel-tetramerization domain containing 13) is regulated by ubiquitination and is involved in the regulation of apoptosis and neural precursor cell proliferation and brain growth[12]. *Kctd13*-deficient mouse neurons display reduced synaptic transmission due to increased RhoA (Ras homolog gene family, member A) expression[13]. Also, induced pluripotent stem cell (iPSC)-derived cortical neurons with CRISPR (Clustered Regularly Interspaced Short Palindromic Repeats)-Cas9 induced loss of *KCTD13* display deficits in neurite formation and network activity[14]. While these studies indicate that the KCTD13 pathway may have a central role in the formation of functional cortical neuron networks in 16p11.2 disorders like autism and SCZ, its effects on development of other neuron subtypes have not been studied in detail.

Brain imaging studies have shown that patients with 16pdel have larger head size[15] and increased body mass index (BMI)[16], and opposite phenotypes have been detected in the patients with 16pdup with decreased head circumference and reduced BMI[4]. Patients with 16pdel also have prominent changes in the brain areas regulating language development—middle, superior temporal gyrus, and caudate, and these deficits are associated with autism diagnosis[17]. Brain imaging studies have also discovered significant and opposite changes in the reward circuitry, including striatum, orbito-frontal cortex, medio-dorsal thalamus, and insula in the patients with 16pdel and duplication, potentially contributing in part to the opposite changes in the body mass development[17–20]. In addition to BMI changes, striatal deficits and abnormal dopaminergic (DA) neuron function are also present in several neuropsychiatric disorders, including SCZ, bipolar disorder, depression, anxiety, and autism[21–24]. DA neurons have a crucial role in the function and development of neural connectivity in the striatal circuitry, and they regulate mood, reward sensation, learning, and several aspects of behavior, including drug and alcohol abuse via dopamine secretion[25,26]. For the treatment of psychiatric disorders and behavioral problems associated with autism, several antipsychotic medications, such as haloperidol or risperidone, are used to target the dopamine pathway and its receptors[27–29]. However, these medications only provide symptomatic treatment for these disorders, and they can cause severe side effects.

Currently, the effects of the 16p11.2 CNV on the striatal function and development of the disease phenotypes in DA neurons and their networks remain largely unknown. Characterization of these cell-type-specific phenotypes is important for the development of new therapies for these disorders and for screening of new compounds that affect DA neurons and their functionality. To address this need, here we established a human DA neuron cell culture model of 16p11.2 CNV. We investigated the functional and genetic characteristics of the DA neurons derived from isogenic human iPSC lines with CRISPR-Cas9-induced 16p11.2 CNVs[30]. Our findings indicate that the KCTD13 pathway and Ras homolog family member A (RHOA) activation have significant roles in the hyperactivation of DA neuron networks in neuropsychiatric disorders with 16pdel, and the inhibition of this signaling pathway can serve as a therapeutic target for these disorders.

## Results

**Differentiation of human iPSCs with 16p11.2 CNVs into DA neurons.** To study disease phenotypes of the 16p11.2 CNVs on midbrain DA neuron development in vitro, we differentiated isogenic CRISPR-Cas9-edited human iPSCs with 16pdel and 16pdup[30] into DA neurons using our previously established protocol with minor modifications[31] (Fig. 1a). For characterization of the neural cells, we used real-time quantitative PCR (qRT-PCR) and immunocytochemistry (ICC). For functional phenotyping of DA neurons, we used patch-clamping, high-density microelectrode array (HD-MEA) recordings, and low-density MEA (LD-MEA) recordings, and for gene expression profiling, we performed RNA sequencing (RNAseq) with the Illumina next-generation sequencing platform (Fig. 1b). Schematic presentation of the study design, DA neuron differentiation, and sample collection time points are shown Fig. 1a, b.

First, we characterized the expression of pluripotency markers OCT4 (octamer binding transcription factor 4), NANOG (Nanog homeobox), and TRA1-60 (podocalyxin) in the undifferentiated pluripotent iPSC colonies (Supplementary Fig. 1) and verified their karyotype (Supplementary Fig. 1). The single-nucleotide polymorphism (SNP) array data showed no significant genomic anomalies between the naive iPSC line (GM08330 control) and the CRISPR-cas9 edited iPSC clones (Supplementary Table 1a), consistent with the previous study and array comparative genomic hybridization (aCGH) data[30] (Supplementary Table 1b). We also verified the expression of genes from the 16p11.2 gene region in the undifferentiated iPSCs and neural precursor cells (NPCs) with qRT-PCR, and these genes were downregulated in the 16pdel cells and upregulated in the 16pdup cells (Supplementary Fig. 1c).

The expression of pluripotency marker genes *OCT4* and *NANOG* were downregulated during the 27-day neuronal differentiation process (Fig. 1c). At the same time, floor-plate patterning factor *FOXA2* (Forkhead Box A2) expression was upregulated in the cells, as well as the expression of the DA neuron markers *EN1* (Engrailed Homeobox 1) and *TH* (Tyrosine Hydroxylase), which were highly expressed at day 27 of differentiation (Fig. 1c). We also detected that >70% of these neural precursor cells were FOXA2 positive at day 5, which confirmed that the floor-plate patterning of the cells with SAG

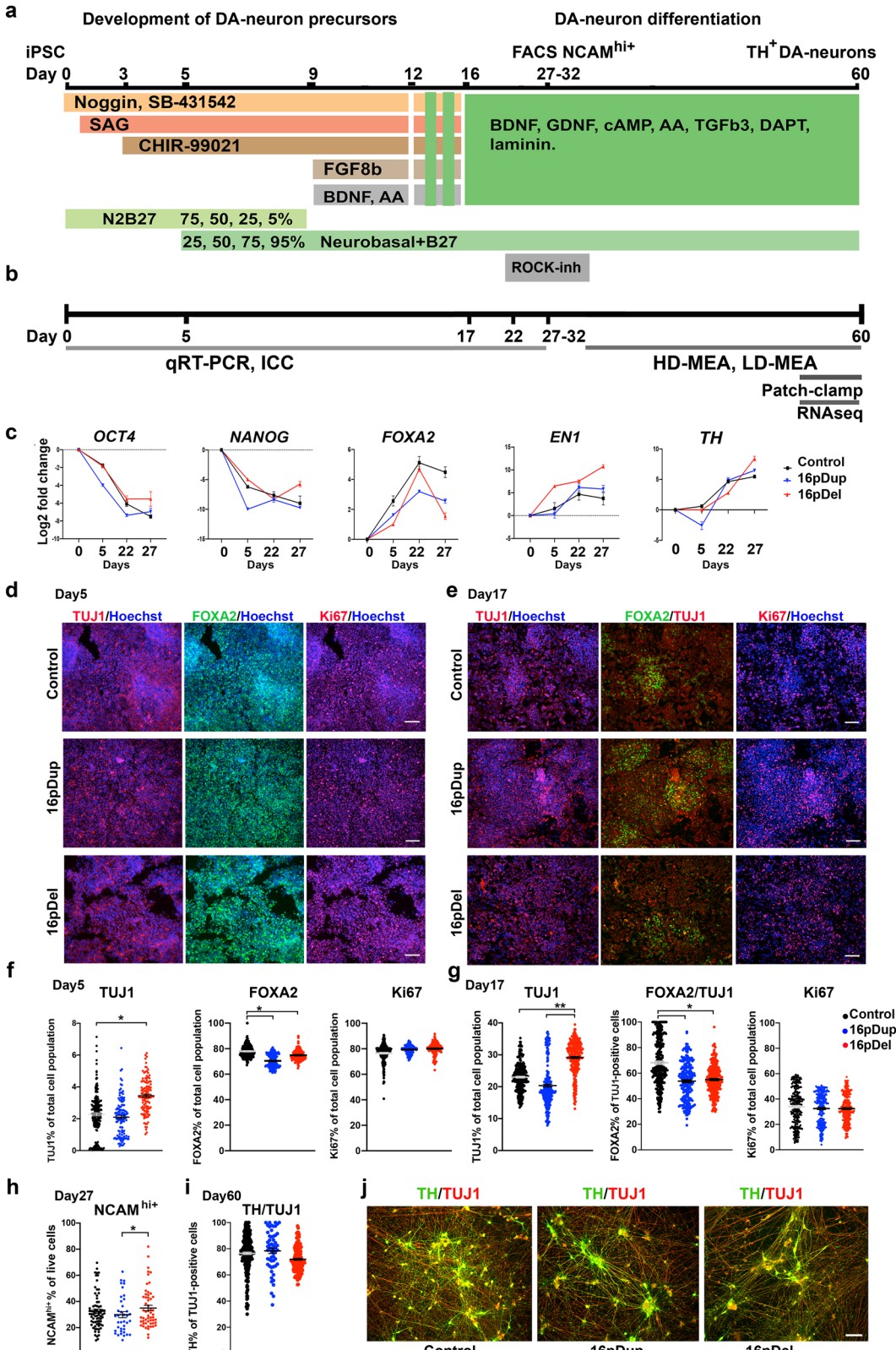

(smoothened agonist) was efficient. Control cells expressed a higher level of FOXA2 compared to cells with 16p11.2 CNVs (Fig. 1d, f). At this stage of the differentiation, the number of TUJ1 (beta-tubulin III)-positive neurons was still low (2–3% from the total cell population), and the 16pdel cells had a significantly higher number of TUJ1-positive cells compared to the control cell population ($p < 0.05$). At day 5 of differentiation, the cell

populations were also highly proliferative with >70% of the cells being Ki67 positive in all the different genotypes studied (Fig. 1d, f).

After the initial neuronal patterning, at day 17 the cells were 20–29% TUJ1 positive, and of these neurons, 54–68% were FOXA2 positive (Fig. 1e, g). Interestingly, the cells with 16pdel expressed significantly higher amounts of TUJ1 compared to

**Fig. 1 Differentiation of human iPSCs with 16p11.2 CNVs into dopaminergic neurons. a** Schematic presentation of the DA neuron differentiation protocol with media supplements, growth factors, and small molecules. **b** Schematic presentation of the study design and sample collection time points. **c** Quantitative RT-PCR for characterization of pluripotency markers OCT4 and NANOG, floor-plate marker FOXA2, and DA neuron markers EN1 and TH. *X*-axis shows days of differentiation. Data are presented as mean ± SD, *n* = 3 independent experiments. **d** Representative images of immunocytochemical characterization of the neural precursor cells at day 5 of differentiation with TUJ1, FOXA2, and Ki67. Scale bars: 33 μm. The experiment was repeated three independent times with similar results. **e** Representative immunocytochemical characterization at day 17 of differentiation with TUJ1, TUJ1/FOXA2, and Ki67. Scale bars: 34 μm. The experiment was repeated six independent times with similar results. **f** Quantification of the TUJ1-, FOXA2-, and Ki67-positive cells of total cell population on day 5. TUJ1 expression was higher in the 16pdel cells compared to control cells (*p* = 0.0231). FOXA2 expression in TUJ1 cells was higher in the control cells compared to 16pdup (*p* = 0.0457) and 16pdel cells (*p* = 0.0308). The number of biologically independent experiments is *n* = 3 per genotype. **g** Quantification of the TUJ1-, FOXA2 positive of TUJ1-, and Ki67-positive cells of total cell population on day 17. TUJ1 expression was higher in the 16pdel cells compared to control cells (*p* = 0.0051) or 16pdup cells (*p* = 0.0046). Expression of FOXA2/TUJ1-positive cells was higher in the control cells compared to 16pdup (*p* = 0.0457) and 16pdel cells (*p* = 0.0379). The number of biologically independent experiments is *n* = 6 per genotype. Data are presented as mean ± SEM, unpaired *t* test with Welch's correction and one-tailed *p* values, *\*p* < 0.05, *\*\*p* < 0.01 in (**f**, **g**). **h** FACS-based quantification of NCAM$^{hi+}$ cells prior cell sorting at day 27 of differentiation. 16pdel cells had higher NCAM$^{hi+}$ expression compared to 16pdup cells (*p* = 0.0386). The number of biologically independent samples are *n* = 74 control, *n* = 38 16pdup, and *n* = 56 16pdel. Unpaired Mann–Whitney test with one-tailed *p* values, *\*p* < 0.05. **i** Quantification of TH-positive cells from TUJ1-positive cell population after sorting and subculturing until total of day 60 of differentiation. The number of biologically independent experiments are *n* = 6 control, *n* = 4 16pdel, and *n* = 3 16pdup. Data are presented as mean ± SEM in (**h**, **i**). **j** Representative images of TH/TUJ1-positive cells at day 60 of differentiation. Scale bars: 85 μm. The experiment was repeated three independent times with similar results. Source data are provided as a Source Data file.

control and 16pdup cells (*p* < 0.05, Fig. 1g). Also, the number of TUJ1-positive neurons co-expressing FOXA2 was lower in the 16pdel cultures compared to control cultures (Fig. 1g), indicating that 16p11.2 CNV caused deficits in the patterning of these cells. At this stage of differentiation, the number of proliferative Ki67-positive precursors was decreased compared to day 5 (Fig. 1f, g), but no differences in the number of proliferative cells were detected between different genotypes.

The small molecule-based neural differentiation protocols that rely on the dual-SMAD inhibition usually result in heterogeneous neuronal cell populations[32,33]. In addition, the 16p11.2 CNVs altered the initial neuronal patterning of these cells showing decreased numbers of neurons in the 16pdup population (Fig. 1d–g). To overcome the heterogeneous nature of stem-cell-derived neuronal cultures, we used NCAM$^{high+}$ (neural cell adhesion molecule) sorting to enrich mature neurons and to increase TH expression in the population, as we previously described[31]. The neuronal cells were sorted after 27–32 days of differentiation, and prior to the cell sorting, the cells were ~20–50% NCAM$^{high+}$ positive (Fig. 1h). After sorting, NCAM$^{high+}$ purity in each population was 95.05 ± 4.41% in control, 94.85 ± 4.28% in 16pdup, and 93.80 ± 3.87% in 16pdel (data are presented as average ± SD; Supplementary Fig. 2n). After 4 weeks of subculturing of the NCAM$^{high+}$ cells, ~72–78% of the cells were TH and TUJ1 positive in these cultures (Fig. 1i, j and Supplementary Fig. 3c), and no difference in the number of TH/TUJ1-positive cells was detected between different genotypes. To verify the ventral midbrain identity of the DA neurons, we quantified TH/NURR1 (nuclear receptor-related 1) expression at day 60; 62.9 ± 10% control, 62.9 ± 9.4% 16pdup, and 62.4 ± 12.3% in 16pdel, data are presented as average ± SD. We also detected co-expression of LMX1A (Lim homeobox transcription factor 1 alpha)/FOXA2 in these cell populations (Supplementary Fig. 3a–c). The number of TUJ1-negative cells were ~15–19% from total cell population. We identified that these cells were SOX2 (SRY-box transcription factor 2)/nestin-positive progenitors; 17.8 ± 11.1% control, 13.1 ± 4.1% 16pdup, and 22.1 ± 9.0% in 16pdel (data are presented as average ± SD, Supplementary Fig 3d). We also detected GABA (gamma-aminobutyric acid)-positive neurons among the neuronal cell populations: 18.9 ± 13.1% in control, 15.1 ± 8.1% in 16pdup, and 11.9 ± 6.9% in 16pdel. No differences between different genotypes were detected in the expression of these markers (Supplementary Fig. 3d). We did not detect any CTIP2-positive cortical layer neurons, or H9B-positive motoneurons, or GFAP-positive (glial fibrillary acidic protein)

astrocytes at day 60 of differentiation (Supplementary Fig. 3j–l). These results indicate that we were able to differentiate DA neurons efficiently from the iPSCs with 16p11.2 CNVs and isogenic controls.

**Transcriptional gene expression analyses reveal overlapping expression of genes associated with autism and SCZ in the DA neurons with 16pdel.** To understand the molecular pathways affecting the development of the cells with 16p11.2 CNVs, we performed molecular characterization and RNAseq for the DA neurons after 50 days of differentiation (Fig. 2). First, we confirmed that the differentiated DA neurons retained the CNV of 16p11.2, by analyzing the expression of eight selected genes from the 16p11.2 region with qRT-PCR (Supplementary Fig. 3i). These genes, *KCTD13*, *CORO1A* (Coronin1A), *TMEM219* (Transmembrane protein 219), *TAOK2*, *CDIPT* (CDP-Diacylglycerol-Inositol 3-phosphatidyltransferase), *MAPK3*, *DOC2A* (Double C2 domain alpha), and *PPP4C* (Protein Phosphatase 4 Catalytic Subunit), were downregulated in the 16pdel cells and upregulated in the 16pdup cells as expected. Then, we verified that all the collected RNA samples with different genotypes expressed comparable levels of DA neuron-specific markers *TH* (Fig. 2a), *LMX1B* (LIM Homeobox Transcription Factor 1Beta), *GIRK2* (G-protein-Activated Inward Rectifier Potassium Channel 2), *AADC* (Aromatic-L-Amino Acid Decarboxylase), *NURR1* (Supplementary Fig. 3g), and the TH-positive neurons expressed dopamine transporter (DAT, Supplementary Fig. 3h). To characterize the effects of 16p11.2 CNV on the development of the DA neurons, we compared the relative expression levels of the genes located at the 16p11.2 region from the RNAseq data between the control and 16pdel neurons, and between control and 16pdup neurons (Fig. 2b). In the 16pdel cells, the expression of these genes was decreased, while in the 16pdup cells variably increased expression of these genes was detected compared to controls (Fig. 2b). To analyze further the effects of these CNVs, we performed a weighted gene co-expression network analysis and identified groups of co-expressed genes that were related to the copy number of the 16p11.2 region (Fig. 2c, d). We identified two separate modules that were highly correlated with 16p11.2 copy number: Module 19 (M19 correlation with 16p11.2 gene dosage; 0.9342) containing genes that were downregulated in the 16pdel neurons and upregulated in the 16pdup neurons compared to control neurons (Fig. 2c), and Module 25 (M25 correlation with 16p11.2 gene dosage; −0.6839) containing genes that were downregulated in the 16pdup neurons and upregulated in the

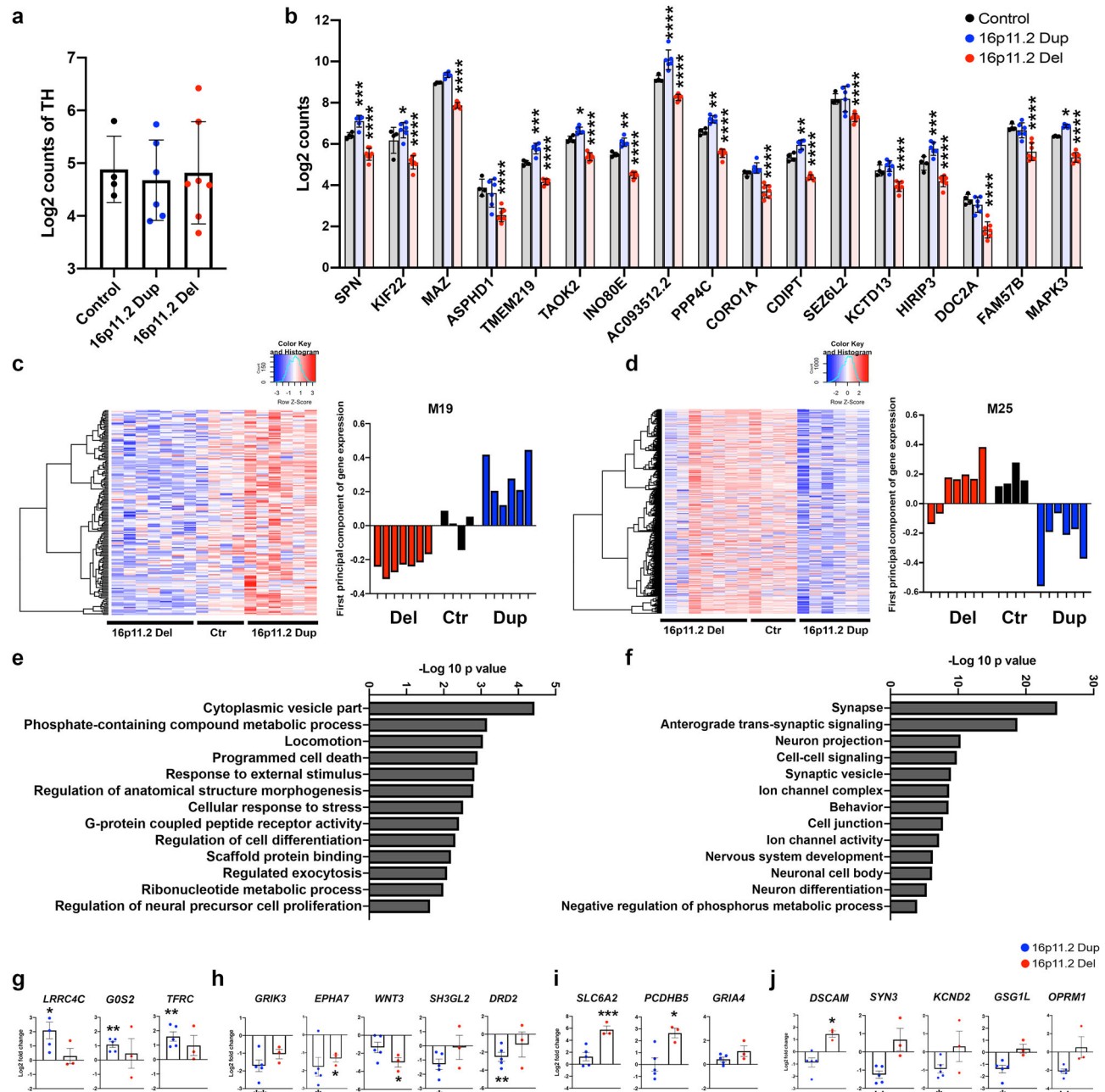

16pdel neurons and control neurons (Fig. 2d). We then performed gene ontology analysis for both of these modules to further understand the functional consequences of these gene expression changes (Fig. 2e, f). Genes within M19 were enriched in functional categories such as cytoplasmic vesicle formation and secretion, positive regulation of phosphorylation and phosphorous metabolic processes, cell death, cellular stress and oxidative stress, ribonucleotide metabolic process, and cell migration and proliferation (Fig. 2e and Supplementary Data 1). In contrast, M25 contained genes that were enriched in categories: synapses, dendrite and axon development, neurotransmitter secretion, ion channel complex activation, regulation of membrane potential, nervous system development, neuronal cell body, and negative regulation of phosphorous metabolic process (Fig. 2f and Supplementary Data 2). In addition, ingenuity pathway analysis (IPA) of these modules revealed that the ERK pathway was central in the gene network of 16pdup neurons in M19 (Supplementary Fig. 4 and Supplementary Data 3), consistent with

previous mouse study[34]. In M25, the synaptic genes were significantly altered in the 16pdel DA neurons, and IPA analyses also revealed that the genes in M25 were associated with disease categories such as (1) cell-to-cell signaling and interaction, nervous system development and function, (2) cancer, dermatological diseases, organismal injury, and abnormalities, (3) neurological disease, (4) psychological disorders, and (5) behavior (Supplementary Fig. 4 and Supplementary Data 4). These findings in 16pdel DA neurons are aligned with the disease categories identified in a previous study of KCTD13-KO iPSC-derived neuronal cells[14].

We also analyzed selected target genes from the M19 and M25 with qRT-PCR to validate whether genes regulating dopamine signaling, neuron development, and drug abuse were altered in DA neurons with 16p11.2 CNVs. In the first group of genes, expressions of LRRC4C, G0S2, and TFRC were significantly higher in 16pdup neurons compared to control cells (Fig. 2g). We also identified interesting genes that were downregulated in

**Fig. 2 Transcriptional gene expression analyses show significant and opposite changes in the neuronal development and differentiation in the DA neurons with 16p11.2 CNVs. a** Expression of *TH* in the control, 16pdup, and 16pdel DA neurons at day 50 of differentiation, analyzed with the next-generation sequencing method (Illumina). **b** Expression of selected genes from 16p11.2 gene region in the control, 16pdup, and 16pdel DA neurons. *SPN* ($p$ = 0.0005 16pdup, $p < 0.0001$ 16pdel), *KIF22* ($p$ = 0.0166 16pdup, $p < 0.0001$ 16pdel), *MAZ* ($p < 0.0001$ 16pdel), *ASPHD1* ($p < 0.0001$ 16pdel), *TMEM219* ($p$ = 0.0005 16pdup, $p < 0.0001$ 16pdel), *TAOK2* ($p$ = 0.0482 16pdup, $p < 0.0001$ 16pdel), *INO80E* ($p$ = 0.0034 16pdup, $p < 0.0001$ 16pdel), *AC093512.2* ($p <$ 0.0001 16pdup, $p < 0.0001$ 16pdel), *PPP4C* ($p$ = 0.0029 16pdup, $p < 0.0001$ 16pdel), *CORO1A* ($p < 0.0001$ 16pdel), *CDIPT* ($p$ = 0.0027 16pdup, $p < 0.0001$ 16pdel), *SEZ6L2* ($p < 0.0001$ 16pdel), *KCTD13* ($p < 0.0001$ 16pdel), *HIRIP3* ($p$ = 0.0004 16pdup, $p < 0.0001$ 16pdel), *DOC2A* ($p < 0.0001$ 16pdel), *FAM57B* ($p < 0.0001$ 16pdel), and *MAPK3* ($p$ = 0.0155 16pdup, $p < 0.0001$ 16pdel). Data are presented as mean ± SD, and the number of biologically independent samples are $n = 4$ control, $n = 6$ 16pdup, and $n = 7$ 16pdel in (**a**, **b**). Gene expression differences were analyzed with two-way ANOVA with Dunnett's multiple comparisons test, *$p < 0.05$, **$p < 0.01$, ***$p < 0.001$, and ****$p < 0.0001$ in (**b**). **c** Gene co-expression module M19 and first-principal component of gene expressions relative to the genes in 16p11.2 region in the three genotypes studied. **d** Gene co-expression module M25 and first-principal component of gene expressions relative to the genes in 16p11.2 region in the three genotypes studied. **e** Gene ontology categories of the module M19 genes that were expressed at lower levels in the 16pdel cells compared to control cells or 16pdup cells. **f** Gene ontology categories of the module M25 genes that were expressed at lower levels in the 16pdup cells compared to control cells or cells with 16p deletion. The number of biologically independent samples in the co-expression analyses are $n = 4$ control, n = 6 16pdup, and $n = 7$ 16pdel in (**c**–**f**). **g**–**j** Quantitative RT-PCR analyses of selected genes associated to DA neuron function in neuropsychiatric disorders and in drug addiction that were detected in the gene co-expression modules and gene ontology categories of RNAseq data. **g** Selected genes that were expressed at higher levels in the 16pdup cells compared to control cells: *LRRC4C* ($p$ = 0.0233 16pdup), *GOS2* ($p$ = 0.0057 16pdup), and *TFRC* ($p$ = 0.0081 16pdup). **h** Selected genes that were expressed at lower levels in the DA neurons with 16.11.2 CNVs compared to control cells: *GRIK3* ($p$ = 0.0066 16pdup), *EPHA7* ($p$ = 0.027 16pdup, $p$ = 0.0227 16pdel), *WNT3* ($p$ = 0.0295 16pdel), *SH3GL2* ($p$ = 0.0237 16pdup), and *DRD2* ($p$ = 0.0074 16pdup). **i** Selected genes expressed at higher levels in the 16pdel cells compared to control cells *SLC6A2* ($p$ = 0.0119 16pdel), *PCDHB5* ($p$ = 0.0217 16pdel), and GRIA4. **j** Selected genes expressed in opposite direction in the 16pdel and 16pdup cells; *DSCAM* ($p$ = 0.0301 16pdel), *SYN3* ($p$ = 0.0036 16pdup), *KCND2* ($p$ = 0.0157 16pdup), *GSG1L* ($p$ = 0.0120 16pdup), *OPRM1* ($p$ = 0.0011 16pdup). **g**–**j** Log 2 fold changes are presented as mean ± SEM compared to gene expression levels of control cells (adjusted to 0), one-sample *t* test and Wilcoxon's test, $p$ values: *$p < 0.05$, **$p < 0.01$, ***$p < 0.001$. The number of biologically independent samples are $n = 3$ control, $n = 5$ 16pdup, and $n = 3$ 16pdel in (**g**–**j**). Source data are provided as a Source Data file.

16pdup neurons compared to control neurons, and variably decreased in 16pdel neurons: *GRIK3*, *EPHA7*, *WNT3*, *SH3GL2*, and *DRD2* (Fig. 2h). In the 16pdel neurons, expression of *SLC6A2*, *PCDHB5*, and *GRIA4* was increased compared to control cells (Fig. 2i). We also found genes that were expressed in opposite directions between 16pdel and 16pdup neurons: *DSCAM*, *SYN3*, *KCND2*, *GSG1L*, and *OPRM1*. These genes were downregulated in the 16pdup group and upregulated in the 16pdel group compared to control cells (Fig. 2i). Triplication of *DSCAM* is associated with neurodevelopmental disorders (e.g., Down syndrome)[35,36], and its increased expression also alters neural connectivity in other cases of intellectual disability[37]. *SYN3* encodes synapsinIII, which is associated with MAPK-ERK pathway activation, synaptogenesis, and SCZ. *KCND2* specifies potassium voltage-gated channel subfamily-D member-2, which affects excitability and action potential (AP) firing in the brain. *GSG1L* makes part of the AMPA receptor complex. *OPRM1* codes for opioid receptor Mu-1 whose reduced expression is associated with drug addiction by regulating dopamine release in the brain during drug or alcohol use[38]. Our findings indicate that expression of these genes predisposing to autism is overlapping with genes associated with SCZ and drug abuse (Supplementary Data 4). Furthermore, these genes are co-expressed during DA neuron network development in the 16pdel neurons, and their expression is decreased in the 16pdup neurons (Supplementary Data 4).

**Synaptic marker expression and synaptic activity is increased in the DA neurons with 16pdel.** Since our gene expression analyses indicated that the 16p11.2 CNVs alter the synaptic marker expression in the cells, next we quantified the expression of synaptic markers on the DA neurons with immunocyto-chemical staining after 60 days of differentiation (Fig. 3). The density of synapsin 1 (SYN1) and postsynaptic density protein 95 (PSD95) puncta on the TUJ1-positive neurites was increased in the 16pdel neurons compared to control neurons ($p < 0.0001$) and compared to duplication neurons ($p < 0.01$, Fig. 3a, b). The expression of synaptophysin1 (SYP1) on the TH-positive DA neurons was increased in the control cells compared to 16pdup

cells ($p < 0.0001$, Fig. 3c, d). The 16pdel neurons also had an increased density of SYP1 puncta compared to 16pdup ($p < 0.0001$) and to control neurons ($p < 0.0001$, Fig. 3c, d). These results indicate that the synapse density is higher in the neurons with 16pdel.

To characterize the functional properties of the DA neurons and their synaptic connections, we assessed excitatory postsy-naptic currents (EPSCs) using patch-clamp recordings (Fig. 3e±g). We detected an increased frequency of EPSCs in 16pdel DA neurons compared to control DA neurons (Fig. 3f, $p < 0.05$). There was no difference in EPSC amplitude between different CNVs or control neurons (Fig. 3g).

**DA neurons with 16pdel display morphological deficits and increased excitability.** To further characterize the morphological and electrophysiological properties of the DA neurons with 16p11.2 CNVs, we performed patch-clamp experiments to mea-sure the spontaneous activity and excitability of the neurons (Fig. 4). To visualize the morphological features of the neurons simultaneously with patch recordings, we included Alexa Fluor dye in the recording pipette and imaged the patched cells (Fig. 4a). We found that soma size was significantly larger in the 16pdel neurons compared to control neurons ($p < 0.01$, Fig. 4b). This finding is consistent with previous data from iPSC-derived cortical neurons with 16pdel compared to control cells[39]. We did not detect significant differences in the number of primary neurites in the 16pdel neurons, but 16pdup neurons displayed an increased number of neurites compared to control cells ($p < 0.01$, Fig. 4b). There was no difference in the length of the longest neurite between CNV and control neurons (Fig. 4b).

Upon entering whole-cell current-clamp (CC) configuration in neurons, we quantified the number of spontaneously active neurons and found that 86% of control cells, 27% of 16pdup cells, and 67% 16pdel cells were spontaneously active (at least one AP observed in 1 min of CC recording at $I_{hold} = 0$ pA). Notably, the number of spontaneously active neurons was significantly lower in 16pdup group compared to control cell population ($\chi^2$ test for control vs. 16pdup, $p < 0.0001$), and compared to 16pdel neurons

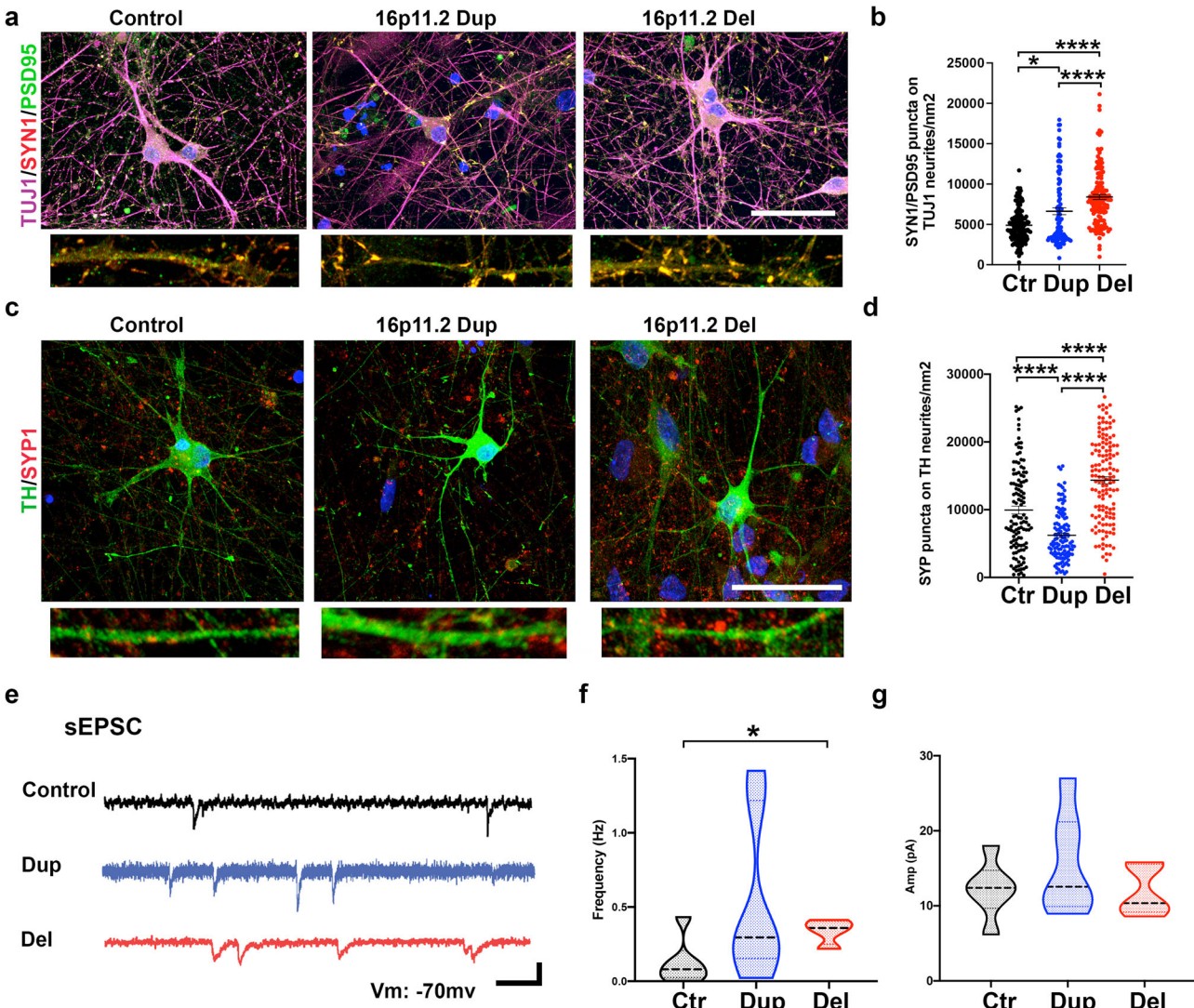

**Fig. 3 16p11.2 deletion caused increased synaptic marker expression and EPSCs in the DA neurons. a** Representative confocal images of the DA neurons stained with TUJ1/PSD95/SYN1. Scale bar 44 μm. The experiment was repeated five independent times with similar results. **b** Quantification of SYN1/PSD95 puncta on TUJ1-positive neurites/nm$^2$. 16pdel cells had higher expression of SYN1/PSD95 compared to control cells ($p < 0.0001$) or compared to 16pdup cells ($p < 0.0001$). Control cells had lower expression of SYN1/PSD95 compared to 16pdup cells ($p = 0.0454$). Data are presented as mean ± SEM, and the number of biologically independent samples is $n = 5$ per each genotype. **c** Representative confocal images of the DA neurons stained with TH and SYP1. Scale bar 44 μm. The experiment was repeated four independent times with similar results. **d** Quantification of SYP1 puncta on TH-positive neurites/ nm$^2$. 16pdel cells had higher expression of SYP1 compared to control cells ($p < 0.0001$) or compared to 16pdup cells ($p < 0.0001$). Control cells had higher expression of SYP1 compared to 16pdup cells ($p < 0.0001$). Data are presented as mean ± SEM, and the number of biologically independent samples is $n = 4$ per each genotype. One-way nonparametric Kruskal–Wallis test was done followed by Dunn's multiple comparisons test and adjusted $p$ values, $p$ values: *$p < 0.05$ and ****$p < 0.0001$ (**b**, **d**). **e** Representative voltage-clamp traces with excitatory activity (EPSCs) from DA neurons (scale bar, 10 pA, 50 ms). **f** Quantification of the EPSC frequency shows a significant increase in the frequency of the 16pdel cells compared to control cells (*$p = 0.0476$). The number of biologically independent cells are $n = 6$ control, $n = 7$ 16pdup, and $n = 4$ 16pdel. **g** Quantification of the EPSC amplitude shows no statistical difference between studied groups of neurons. The number of biologically independent samples are $n = 6$ control, $n = 7$ 16pdup, and $n = 5$ 16pdel. Two-tailed nonparametric Kolmogorov–Smirnov test (**f**, **g**). Source data are provided as a Source Data file.

($\chi^2$ test, $p < 0.01$). No significant difference in the number of spontaneously active neurons was detected between the control and 16pdel neurons ($\chi^2$ test, $p = 0.097$). Quantification of Rheobase, a value that describes the minimum current to evoke an AP from holding potential ($V_m = -60$ mV), revealed increased excitability of the 16pdel neurons ($4.86 \pm 0.74$ pA, $n = 23$) compared to control neurons (Control, $7.91 \pm 0.65$ pA, $n = 26$, $p < 0.02$) and compared to 16pdup neurons (16pdup, $10.4 \pm 1.75$ pA, $n = 23$, $p < 0.01$, Fig. 4d). To further characterize higher AP firing frequencies, we injected a large current pulse (50 pA) and detected significantly higher AP firing frequencies in the

16pdel neurons compared to control neurons (Control, $2.49 \pm 0.37$ APs, $n = 11$ vs. 16pdel, $3.93 \pm 0.60$ APs, $n = 12$, $p < 0.05$, Fig. 4f), which indicated that the 16pdel leads to the development of hyperexcitable DA neurons. To evaluate if the observed shift in excitability was rooted in differences in cell size, we quantified cell capacitance, a direct correlate of surface area. Although the control cells were slightly smaller than the 16pdup or 16pdel neurons, we did not detect significant differences in the capacitance between different genotypes (Control: $23.3 \pm 2.3$ pF, 16pdup $26.5 \pm 1.6$ pF, and 16pdel $25.0 \pm 3.1$ pF; Control vs. 16pdup, $p = 0.27$ and Control vs. 16pdel, $p = 0.66$), suggesting

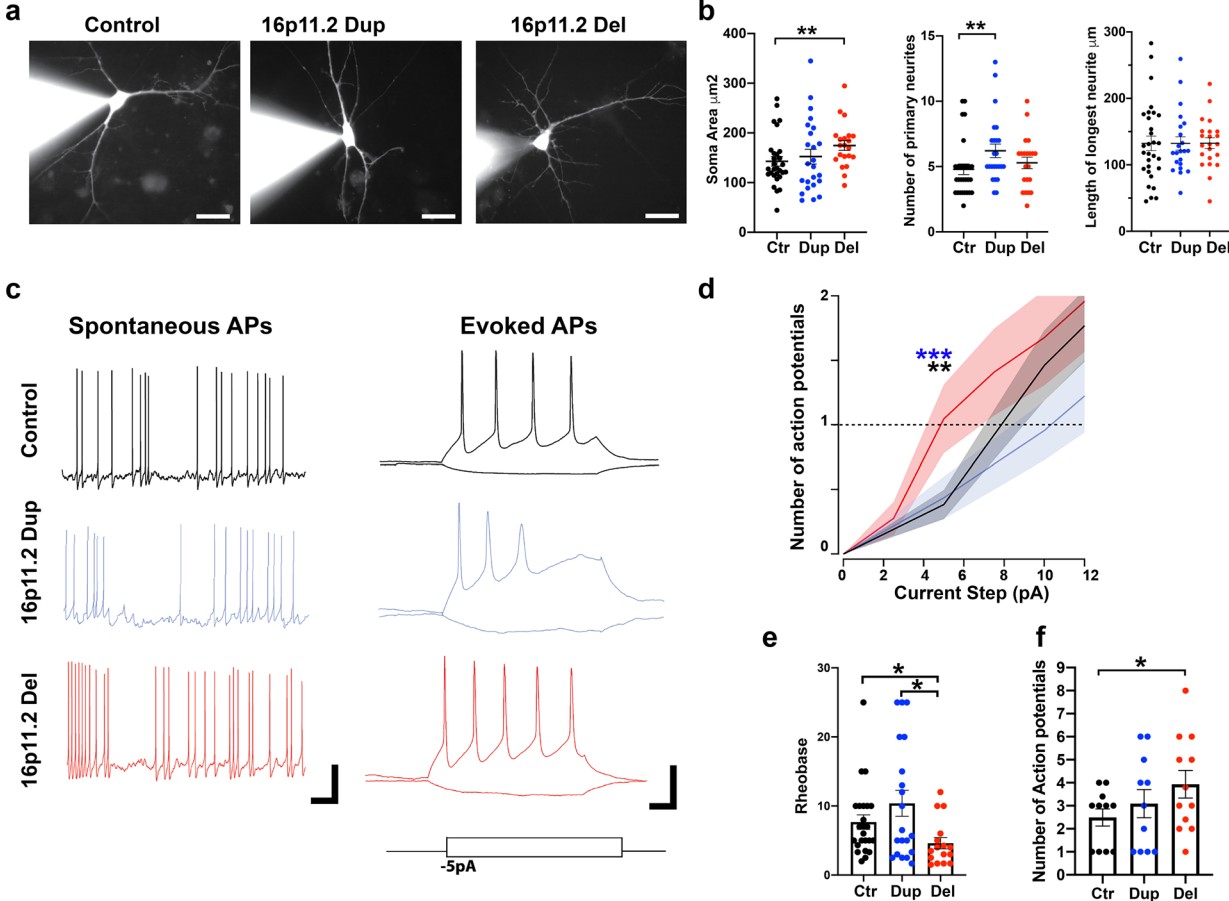

**Fig. 4 16p11.2 deletion caused morphological changes and hyperexcitability of the DA neurons. a** Representative images of neurons loaded with Alexa Fluor 488 dye (1 μM) during recordings. Scale bar 25 μm. The experiment was repeated three independent times with similar results. **b** Characterization of soma size, number of primary neurites, and length of longest neurite of the patched cells. Soma area was significantly increased in the 16pdel cells compared to control cells ($p = 0.0076$). The number of primary neurites was significantly higher in the 16pdup cells compared to control cells ($p = 0.0091$). There was no statistical difference in the length of longest neurite between the cell groups studied, nonparametric Mann–Whitney test with two-tailed $p$ values, **$p < 0.01$. The number of biologically independent samples for soma size and neurite number analyses are $n = 29$ control, $n = 24$ 16pdup, and $n = 21$ 16pdel. The number of biologically independent samples for neurite length analyses are $n = 30$ control, $n = 22$ 16pdup, and $n = 22$ 16pdel. Data are presented as mean ± SEM. **c** Left, representative traces of spontaneous action potentials (APs) at resting membrane ($V_m$) ($I_{hold} = 0$ pA) in the control, 16pdup, and 16pdel cells. $V_m$ of representative current-clamp traces in 16pdel, control, and 16pdup neurons, −38, −39, and −37 mV, respectively. Scale bar 25 mV and 1 s. Right, representative current-clamp recordings demonstrate action potentials in response to current steps of (+15 and −5 pA). Scale bar 10 mV, 50 ms. **d** Excitability analyses of the DA neurons revealed minimum current input (pA) required to generate the first action potential was significantly lower in the 16pdel cells compared to control cells. For generating the input–output excitability curve, control, 16pdel, and 16pdup neurons were current clamped to a $V_m$ potential of −60 mV. 16pdel neurons generated significantly higher number of action potentials compared to control cells or 16pdup cells, **$p < 0.02$ and ***$p < 0.01$. Data are presented as mean ± SEM. **e** Rheobase values (pA) for control, 16pdup, and 16pdel neurons show significantly lower Rheobase for 16pdel cells vs. control cells ($p = 0.0126$), and 16pdel vs. 16pdup cells ($p = 0.0246$), nonparametric Mann–Whitney test with two-tailed $p$ values, *$p < 0.05$. The number of biologically independent samples are $n = 24$ control, $n = 20$ 16pdup, and $n = 16$ 16pdel. **f** Characterization of the number of APs generated by injection of a maximum current stimulation (50 pA) to control, 16pdup and 16pdel neurons. 16pdel neurons generated significantly higher number of APs compared to control cells ($p = 0.028$), parametric unpaired $t$ test with Welch's correction and one-tailed $p$ value, *$p < 0.05$. The number of biologically independent samples are $n = 11$ control, $n = 11$ 16pdup, and $n = 12$ 16pdel. Data are presented as mean ± SEM (**e, f**). Source data are provided as a Source Data file.

that the observed shift in Rheobase is unlikely resulting from an intrinsic property correlated with cell size.

**16pdel leads to hyperactivity of the developing DA neuron networks.** Since our synaptic marker expression analyses showed increased expression of synaptic markers and increased excitability in the neurons with 16pdel, we next asked whether these deficits affect the formation of functional networks during differentiation. For these analyses, we utilized high-density MEA (MaxWell Biosystems) (Figs. 5 and 6 and Supplementary Figs. 5a and 7). The advantage of the HD-MEA platform is that it can

record extracellular network function simultaneously from 1024 sensors selected out of 26,400 sensors per chip yielding very high-resolution data of the neuronal activity during the formation of the network connections[40,41]. To investigate the DA neuron network development in vitro, we recorded the DA neurons over 4 weeks on the HD-MEA chips. During this developmental time, the firing rate was higher in the 16pdel networks compared to control or the 16pdup networks (at day 28 after plating; 0.93 ± 0.2 Hz 16pdel vs. 0.26 ± 0.07 Hz ctr vs. 0.24 ± 0.03 Hz 16pdup, $p < 0.0001$, Fig. 5b). Next, we investigated the propensity of the DA neurons to fire APs in a synchronous matter since synchronization of neuronal firing is important for the DA neuron

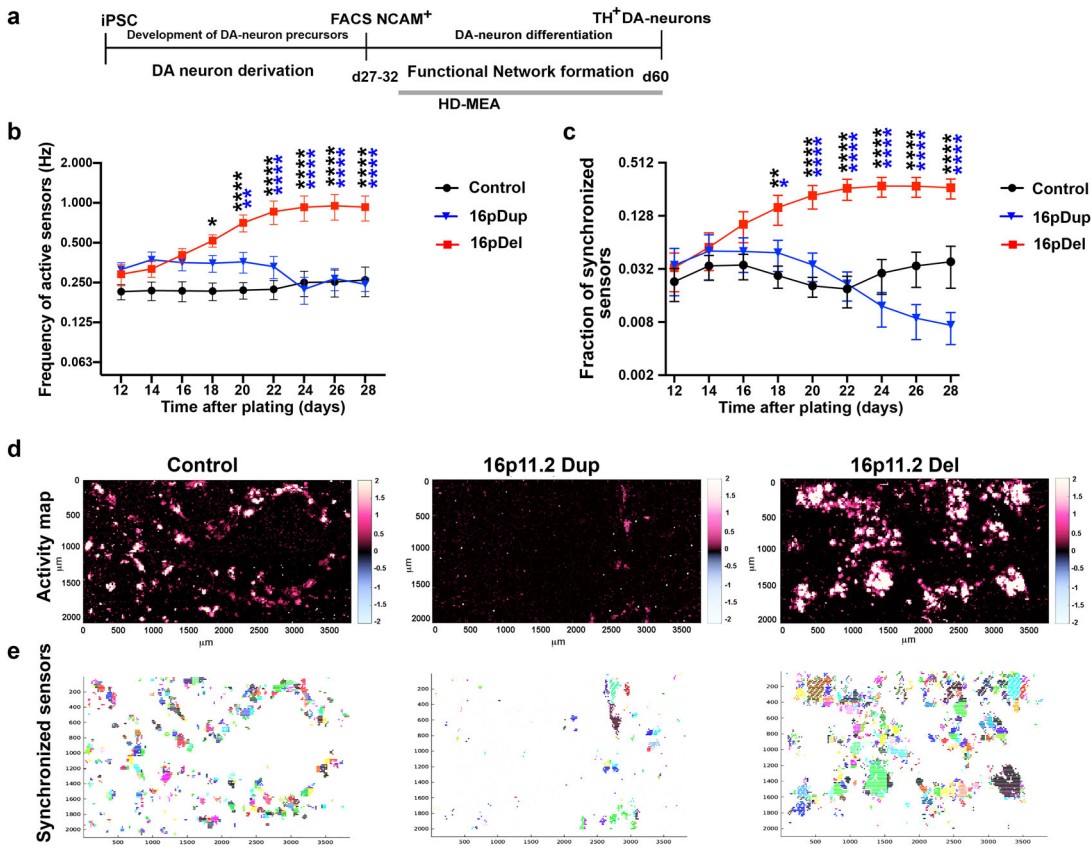

**Fig. 5 16p11.2 deletion caused development of hyperactive DA neuron networks in vitro. a** Schematic representation of the timeline for the functional network analyses. Cells were plated to HD-MEA after sorting and recorded for the 28 days. **b** Frequency of active sensors (Hz) of the control, 16pdup, and 16pdel cells. Significantly higher frequency of the network activity was detected in the 16pdel neurons compared to control or 16pdup groups. Adjusted $p$ values for control vs. 16pdel: $p = 0.017$ d18, $p < 0.0001$ d20–d28, for 16pdel vs. 16pdup: $p = 0.0036$ d20, $p < 0.0001$ d22–28. **c** Fraction of synchronized sensors in the control, 16pdup, and deletion groups. Significantly higher amount of sensors was synchronized in the 16pdel network compared to control or 16pdup groups. Adjusted $p$ values for control vs. 16pdel: $p = 0.0034$ d18, $p < 0.0001$ d20–d28, for 16pdel vs. 16pdup: $p = 0.0141$ d18, $p < 0.0001$ d20–28. Datasets were calculated by making interpolation of each data point recorded per chip per day, and averaged data from each group of chips together based on their genotype: control ($n = 6$), 16pdup ($n = 7$), and 16pdel ($n = 4$) (**b, c**). Data are presented as mean ± SEM. Two-way ANOVA followed by Tukey's multiple comparison test, and adjusted $p$ values are presented, $*p < 0.05$, $**p < 0.01$, and $****p < 0.0001$ (**b, c**). **d** Representative activity maps of the control, 16pdup, and 16pdel DA neurons after 28 days of recording on HD-MEA. **e** Representative figures of the synchronized regions in the DA neuron networks of the control, 16pdup, and 16pdel groups. Clusters of sensors with the same color represent the areas that have synchronized activity within the subarray recording. Source data are provided as a Source Data file.

network formation and bursting capacity[42]. The fraction of the synchronized sensors over all sensors increased significantly over time in the 16pdel networks compared to control networks (at day 28 after plating; $0.27 \pm 0.07$ 16pdel vs. $0.04 \pm 0.019$ ctr, $p < 0.0001$) or compared to 16pdup neuronal networks (at day 28 after plating; $0.27 \pm 0.07$ 16pdel vs. $0.01 \pm 0.003$ 16pdup, $p < 0.0001$, Fig. 5c).

Synchronous bursting of DA neurons is known to regulate DA secretion, and in the ventral tegmental area, this activity is known to cause slow oscillation patterns[42]. Abnormal bursting of DA neurons has been described in patients with obsessive-compulsive disorder (OCD) and SCZ. Thus, to characterize the effects of 16p11.2 CNVs on the development of the bursting patterns in the iPSC-derived DA neuron networks, we utilized a previously developed algorithm for burst detection[43] to analyze human iPSC-derived neuronal networks on HD-MEA. First, we analyzed the number of bursts per minute per all sensors. After 20 days of plating the neurons on the HD-MEA, the 16pdel neurons had a significantly increased number of bursts compared to control cells ($0.386 \pm 0.134$ standard error of the mean (SEM) 16pdel vs. $0.017 \pm 0.009$ SEM ctr, $p < 0.0001$) or compared to 16pdup neurons ($0.023 \pm 0.008$ SEM 16pdup, $p < 0.0001$) (Fig. 6a). Then, we

analyzed the number of spikes per burst and found an increased number of spikes in the 16pdel DA neurons compared to control cells ($6.24 \pm 0.49$ SEM 16pdel vs. $4.29 \pm 0.24$ SEM ctr, $p < 0.05$) or compared to 16pdup neurons ($5.34 \pm 0.70$ SEM), after 20 days of plating on the HD-MEA (Fig. 6b). In addition, we analyzed the duration of the bursts in these neuronal networks. Although no significant difference between 16pdel and control cells was detected, there was a trend towards increased burst duration in the 16pdel neurons (Fig. 6c). Moreover, the 16pdup cells had shorter burst durations compared to 16pdel neurons after 14 days of plating on HD-MEA (Fig. 6c). Finally, the interval between bursts was shorter in the 16pdel neurons compared to control cells ($4.2 \pm 0.9$ s 16pdel vs. $7.0 \pm 0.75$ s ctr, $p < 0.01$) or compared to 16pdup cells ($6.2 \pm 0.7$ s 16pdup, $p < 0.05$, Fig. 6d) on day 20. These data indicate that the DA neuron networks with 16pdel were bursting at higher levels and more frequently than the control or 16pdup networks (Fig. 6d). Representative raster plots of the bursting patterns of the sensors with highest burst rate (Fig. 6e) and the sensors with most spikes/bursts (Fig. 6f) in the DA neuron networks from control, 16pdup, and 16pdel neurons are shown after 20 days of plating on the HD-MEA (Fig. 6e, f).

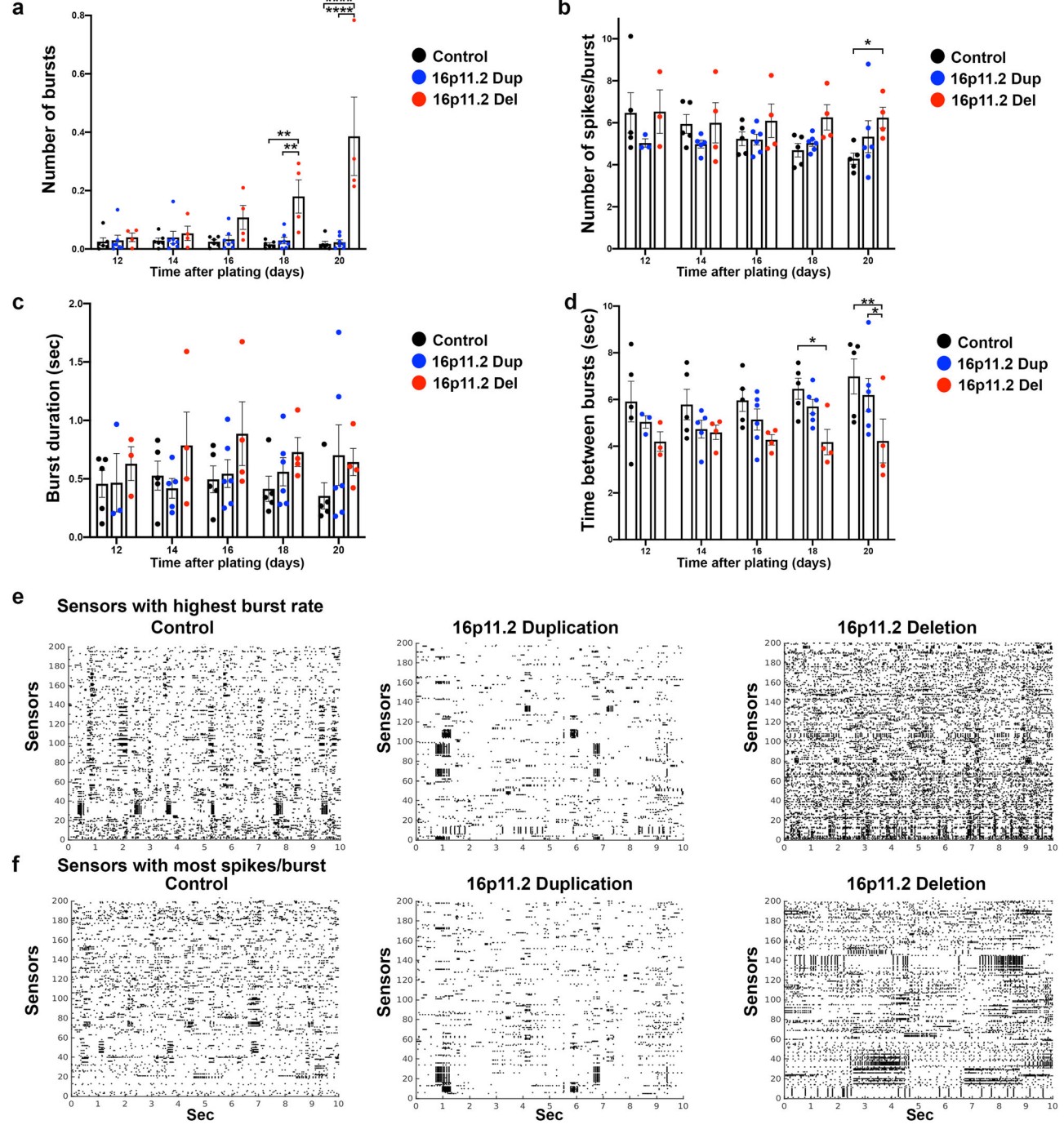

**Fig. 6 DA neuron networks with 16p11.2 deletion have increased bursting phenotypes. a** Quantification of the numbers of bursts/min over all sensors. 16pdel neurons have higher number of bursts compared to control or 16pdup neurons, adjusted $p$ values for control vs. 16pdel: $p = 0.0016$ d18, $p < 0.0001$ d20 and for 16pdel vs.16pdup: $p = 0.0029$ d18, $p < 0.0001$ d20. **b** Quantification of the number of spikes/burst. Increased spike number/burst was detected in the 16pdel neurons compared to control neurons after 20 days of plating on HD-MEA, adjusted $p$ values for control vs. 16pdel: $p = 0.031$ d20. **c** Quantification of the burst duration (s). No significant differences were detected in the burst durations between control, 16pdup, or 16pdel neurons. **d** Quantification of the time between bursts (s). Shorter time between bursts were detected in the 16pdel neurons compared to control or 16pdup neurons, adjusted $p$ values for control vs. 16pdel: $p = 0.0237$ d18, $p = 0.0049$ d20, and for 16pdel vs. 16pdup: $p = 0.0471$ d20. **e** Representative raster plots of 200 selected sensors with highest burst rate/chip from the control, 16pdup, and 16pdel groups. **f** Representative raster plots of 200 selected sensors with most spikes per burst/chip from the control, 16pdup, and 16pdel groups. Datasets were calculated by making interpolation of each data point recorded per chip per day, and averaged data from each group of chips together based on their genotype: control ($n = 6$), 16pdup ($n = 7$), and 16pdel ($n = 4$). Data are presented as mean ± SEM (**a–d**). Two-way ANOVA followed by Tukey's multiple comparisons test, and adjusted $p$ values, presented $p$ value *$p < 0.05$, **$p < 0.01$, and ****$p < 0.0001$. Source data are provided as a Source Data file.

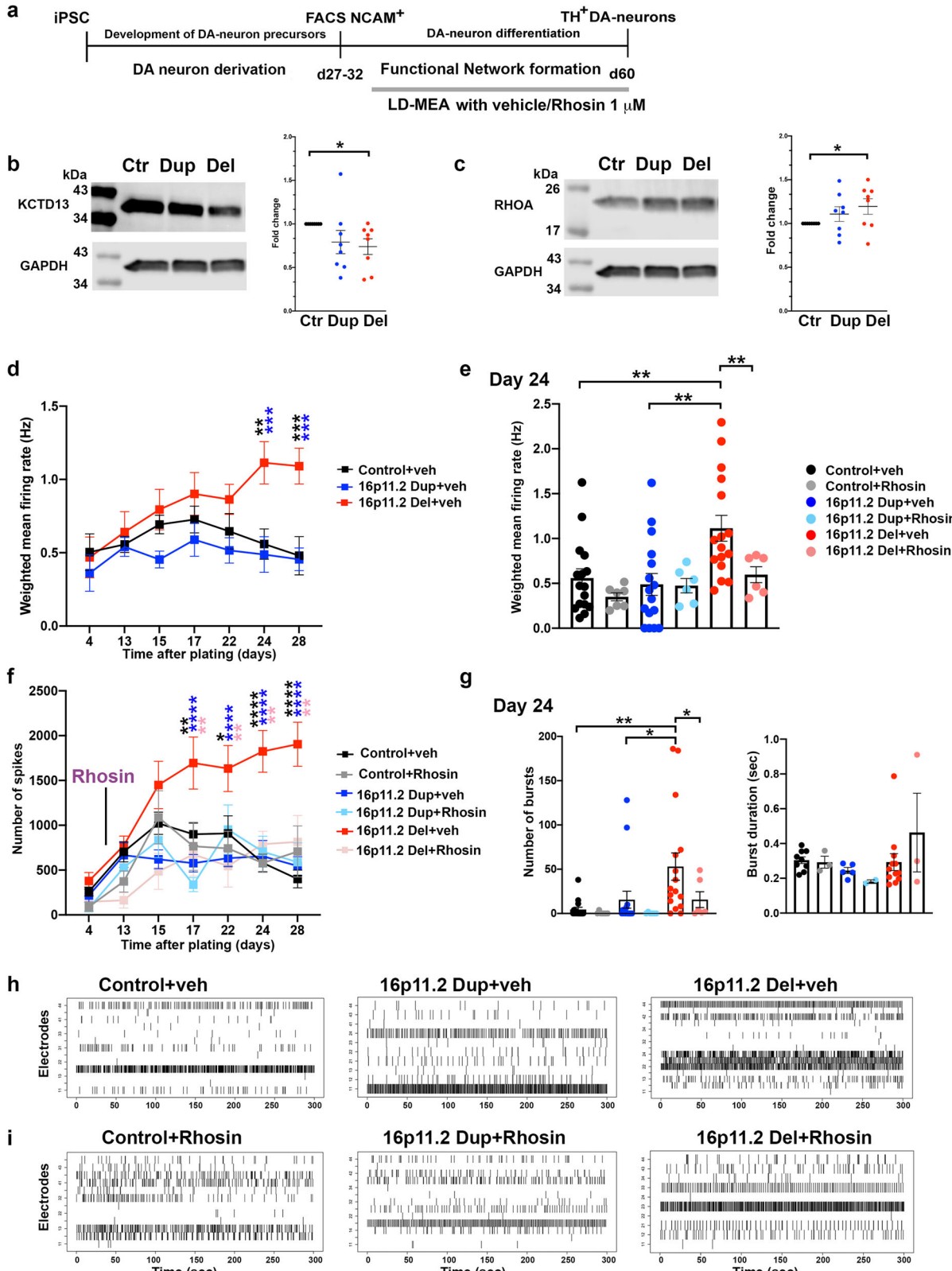

**Increased RHOA expression in the DA neurons with 16pdel.** Previous studies have shown that *KCTD13* expression plays an important role in the functional development of cortical neurons[13,14,44]. Thus, to analyze the molecular pathway involved in the functional deficits in the DA neuron networks with 16p11.2 CNVs, we analyzed the expression of the KCTD13 by western blot (Fig. 7b). In the 16pdel cells, expression of KCTD13 was

reduced compared to control cells ($p < 0.05$, Fig. 7b). At the molecular level, KCTD13 interacts with the Cullin3–ubiquitin complex, which regulates activation of RHOA[13,44]. RHOA expression was also increased in the 16pdel cells compared to control cells ($p < 0.05$, Fig. 7c). Although we detected a trend towards increased RHOA expression in the 16pdup neurons compared to controls, this change was not statistically significant

**Fig. 7 Rhosin treatment rescued functional deficits of the 16p11.2 deletion DA neuron networks in vitro. a** Schematic presentation of the network functional analyses and Rhosin (1 μM) or vehicle treatment on the LD-MEA. Cells were plated to LD-MEA after sorting and recorded up to 28 days. **b** *KCTD13* expression in the DA neurons with 16p11.2 CNVs. Reduced *KCTD13* expression was detected in the 16pdel neurons compared to control cells ($p = 0.0117$). **c** RHOA expression in the DA neurons with 16p11.2 CNVs. Increased RHOA expression was detected in the 16pdel neurons compared to control cells ($p = 0.0323$). Data are presented as mean ± SEM, and statistical analyses were done with unpaired *t* test with Welch's correction and with one-tailed *p* values, \**p* < 0.05 and \*\**p* < 0.01 (**b, c**). The number of biologically independent samples are n = 8 control, n = 8 16pdup, and n = 8 16pdel (**b, c**). **d** Weighted mean firing rate (Hz) of the active electrodes in DA neuron networks were analyzed during 28 days of recording. Higher weighted mean firing rates were detected in 16pdel neurons compared to control neurons at days 24 and 28 ($p = 0.0015$ day 24, $p = 0.0004$ day 28), and compared to 16dup cells at days 24 and 28 ($p = 0.0003$ d24, $p = 0.0002$ day 28). The number of biologically independent samples is n = 16 per each genotype. **e** After 24 days of plating the DA neurons on the LD-MEA, the weighted mean firing rate (Hz) of 16pdel cells was higher compared to control cells ($p = 0.0021$) or 16pdup cells ($p = 0.0012$), and the long-term Rhosin treatment reduced the weighted mean firing rate in the 16pdel neurons significantly ($p = 0.0032$). **f** The number of spikes were recorded during development of networks up to 28 days. Rhosin or vehicle treatments were started after 5 days of plating of the cells on the LD-MEA. 16pdel cells had higher spike rate compared to control neurons at day 17 ($p = 0.0038$), day 22 ($p = 0.0117$), day 24 ($p < 0.0001$), and day 28 ($p < 0.0001$), and compared to 16dup cells at day 17 ($p < 0.0001$), day 22 ($p < 0.0001$), day 24 ($p < 0.0001$), and day 28 ($p < 0.0001$). Rhosin treatment reduced the spike rate in the 16pdel neurons significantly at day 17 ($p = 0.0073$), day 22 ($p = 0.0032$), day 24 ($p = 0.0061$), and day 28 ($p = 0.0032$). Data presented as mean ± SEM (**d–f**). The number of biologically independent samples are n = 16 for control + veh, 16pdup + veh, 16pdel + veh, and n = 7 control + Rhosin, n = 6 16pdup + Rhosin, n = 6 16pdel + Rhosin (**e, f**). Two-way ANOVA followed by Tukey's multiple comparisons test, and adjusted *p* values, \**p* < 0.05, \*\**p* < 0.01, \*\*\**p* < 0.001, and \*\*\*\**p* < 0.0001 in (**d, f**), and unpaired *t* test with Welch's correction and one-tailed *p* values (**e**). **g** After 24 days of plating the DA neurons, the number of bursts was higher in the 16pdel group compared to control ($p = 0.0035$) or 16pdup cells ($p = 0.025$), and the long-term Rhosin treatment reduced the number of bursts significantly in the 16pdel neurons ($p = 0.0247$). Data are presented as mean ± SEM, unpaired *t* test with Welch's correction and one-tailed *p* values, p values presented \**p* < 0.05 and \*\**p* < 0.01. The number of biologically independent samples are n = 16 for control + veh, 16pdup + veh, 16pdel + veh, and n = 7 control + Rhosin, n = 6 16pdup + Rhosin, and n = 6 16pdel + Rhosin (**g**). The burst duration was not significantly different between different groups. The number of samples per group are control + veh n = 9, 16pdup + veh n = 5, 16pdel + veh n = 12, control + Rhosin n = 3, 16pdup + Rhosin n = 2, 16pdel + Rhosin n = 3 (**g**). **h, i** Representative raster plots of DA neuron network firing and bursting patterns after 24 days of recording on LD-MEA, cells were treated with vehicle or Rhosin. Source data are provided as a Source Data file.

(Figs. 7c and 8b). We also measured Cullin3 expression in these cell populations and detected increased expression in the 16pdel neurons compared to controls (Fig. 8a, b), which indicates that reduction of KCTD13 plays a key role in the KCTD13–Cullin3 complex that facilitates the increase of RHOA pathway activation in the 16pdel neurons. Previous studies had reported that heterozygous loss of gene expression in cells with deletion of CNV usually leads to more prominent changes in protein expression and disease severity compared to cells with duplication of CNV due to compensatory protein expression mechanisms in the duplication cells[45,46]. In line with this, we did not detect significant changes in the protein expression in the 16pdup neurons compared to control cells (Fig. 7b, c).

**Long-term Rhosin treatment rescues the hyperactivation of the 16pdel DA neuron networks.** For functional characterization of spontaneous network activity and effects of drug treatments, we used multi-well LD-MEA recordings with CytoView-MEA plates (Axion BioSystems), which contain 48-wells with 16 electrodes per well. This platform allowed us to do simultaneous recordings from multiple wells, and facilitated recordings from multiple cell lines and treatment conditions in parallel. First, we confirmed that the functional phenotypes of the networks were in line with our data obtained from the HD-MEA platform (Figs. 5 and 6). The functional disease phenotypes of the 16pdel cells were consistent between HD-MEA and LD-MEA recording platforms, as shown by the calculated weighted mean firing rates of the networks with 16p11.2 CNVs neurons compared to control neurons (Fig. 7d). After 24 and 28 days of plating of the cells on LD-MEA, the 16pdel neurons had significantly increased firing frequency compared to control cells or 16pdup cells; day 24: 1.14 ± 0.14 Hz 16pdel vs. 0.56 ± 0.10 Hz ctr ($p < 0.01$), and 1.14 ± 0.14 Hz 16pdel vs. 0.49 ± 0.12 Hz 16pdup ($p < 0.001$). Day 28: 1.09 ± 0.12 Hz 16pdel vs. 0.48 ± 0.13 Hz ctr ($p < 0.001$) and 1.09 ± 0.12 Hz 16pdel vs. 0.45 ± 0.08 Hz 16pdup ($p < 0.001$, Fig. 7d).

To inhibit increased RHOA in the 16pdel neurons, we treated the cells with specific RHO-GTPase inhibitor Rhosin (Tocris) at 1 μM during long-term functional analyses of the DA neuron

network development on the LD-MEA. Rhosin was added for the neurons 5 days after plating of the cells on the LD-MEA and was maintained in the culture media until the end of the recordings (Fig. 7a). Since there was a significant difference in the weighted mean firing rate between control cells and 16pdel cells after 24 days of plating, we asked whether at this time point the Rhosin treatment could rescue the increased network activity of the 16pdel neurons (Fig. 7e). Rhosin treatment decreased significantly the weighted mean firing rate of the 16pdel DA neuron networks compared to vehicle-treated 16pdel neurons (1.14 ± 0.14 Hz 16pdel + veh vs. 0.60 ± 0.09 16pdel + Rhosin, $p < 0.05$, Fig. 7e).

To further characterize the development of the DA neuron networks, we quantified the number of spikes over time in the networks with 16p11.2 CNVs compared to control cells. Although we detected an increase in the number of spikes over time in the control DA neuron networks at day 4 after plating (259 ± 51 spikes) until day 15 (up to 1022 ± 124 spikes), the spike rate plateaued between days 15 and 22, and then was reduced to 401 ± 99 spikes at day 28. After culturing the neurons over 3 weeks on MEA, the DA neuron networks may be getting more sparse due to neuronal migration and clustering, which may affect the overall activity of the neurons after day 22. However, we detected more frequent spikes in the 16pdel neuron networks compared to other genotypes or treatments starting at 17 days after plating (1694 ± 290 spikes) and lasting until day 28 of the recordings (1904 ± 245 spikes). Treatment of the 16pdel neurons with Rhosin over time reduced significantly the number of spikes in the developing DA neuron network (Fig. 7f, $p < 0.01$). Rhosin treatment did not have significant effects on the number of spikes in the control networks or 16pdup networks (Fig. 7f), implicating that the effect of RHOA in the function of neurons is disproportionately greater when it is expressed beyond normal physiological levels in the cells.

We next studied in more detail the number of bursts with the LD-MEA platform after 24 days of plating of the DA neurons. The 16pdel cells had significantly increased the number of bursts compared to the control cell population (53 ± 15.5 16pdel vs. 4.5 ± 2.5 ctr, $p < 0.01$, Fig. 7g) and compared to 16pdup cells (53 ±

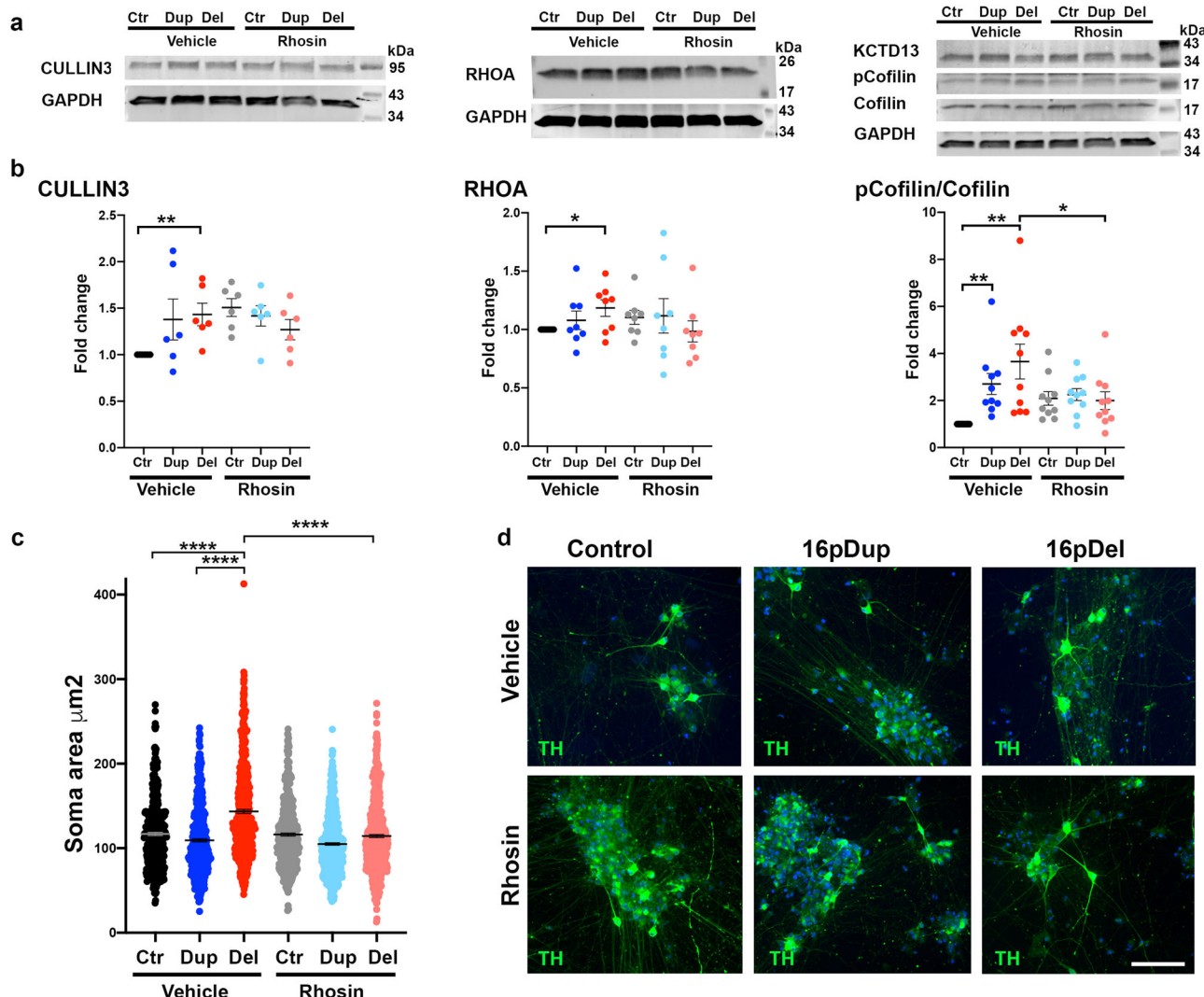

**Fig. 8 Effects of Rhosin treatment on RHOA pathway activation and development of DA neuronal morphology. a** Expression of Cullin3, KCTD13, RHOA, and phospho-Cofilin/Cofilin in DA neurons at day 60 of differentiation. Cells were treated with vehicle (DMSO) or Rhosin 1 μM for 23 days. The experiment was repeated three independent times with similar results. **b** Quantification of Cullin3, RHOA, and phospho-Cofilin/Cofilin. The 16pdel neurons had increased expression of Cullin3 compared to control cells ($p = 0.0079$). RHOA expression was also increased in the 16pdel neurons compared to control cells ($p = 0.0183$). Expression of pCofilin was increased in 16pdel cells compared to control cells ($p = 0.0030$) and in the 16pdup cells compared to control cells ($p = 0.0021$). Rhosin treatment reduced significantly pCofilin expression in the 16pdel cells ($p = 0.0335$). Unpaired $t$ test with Welch's correction and one-tailed $p$ values, *$p < 0.05$ and **$p < 0.01$. The number of biologically independent samples are $n = 6$ per genotype for Cullin3, $n = 8$ per genotype for RHOA, and $n = 10$ per genotype for pCofilin/Cofilin. **c** Soma size measurements of TH-positive DA neurons. 16pdel DA neurons had increased soma area compared to control cells ($p < 0.0001$) or 16pdup cells ($p < 0.0001$). Long-term Rhosin treatment reduced significantly the soma area in the 16pdel cells ($p < 0.0001$). One-way ANOVA and Tukey's multiple comparisons test, ****$p < 0.0001$. The number of independent experiments are $n = 6$ for vehicle treatments per genotype and $n = 6$ for Rhosin treatments per genotype. Data are presented as mean ± SEM in (**b**, **c**). **d** Representative images of TH-positive soma in cell cultures treated with vehicle (DMSO) or Rhosin. Scale bar 85 μm. The experiment was repeated three independent times with similar results. Source data are provided as a Source Data file.

15.5 16pdel vs. 15.6 ± 9.6 16pdup, $p < 0.05$). These data are in line with our findings with the HD-MEA recordings (Fig. 6). In addition, we detected a decreased number of bursts in the 16pdel networks after Rhosin treatment (53 ± 15.5 16pdel + veh vs. 15.7 ± 9.0 16pdel + Rhosin, $p < 0.05$, Fig. 7g). There were no significant differences in the burst duration between 16pdel, control, and 16pdup neurons (Fig. 7g). These data are in line with our burst duration data of the HD-MEA recordings (Fig. 6c). Rhosin treatment did not have any effect on burst duration in the 16pdel neurons (Fig. 7g).

To further characterize the effects of Rhosin in DA neurons, we measured the phosphorylation of cofilin, which is a downstream target for RHOA pathway activation (RHOA–LIMK2–cofilin)[47].

Phospho-cofilin levels were increased in the 16pdel neurons compared to other groups, and Rhosin treatment reduced RHOA and phospho-cofilin in 16pdel neurons (Fig. 8a, b). In addition, we quantified the soma size of TH-positive neurons and detected significantly increased soma area of the 16pdel DA neurons compared to control or 16pdup DA neurons, and Rhosin treatment reversed this phenotype (Fig. 8c, d). Taken together, the inhibition of the specific RHO-GTPase with Rhosin treatment during DA neuron network development rescued the cell size and hyperactivity phenotypes of the 16pdel neurons and reduced spiking and bursting of the neurons, indicating that RHOA inhibition may be a therapeutically relevant target for the neuropsychiatric disorders associated with 16pdel.

## Discussion

The 16p11.2 CNVs are a major genetic risk factor for various neuropsychiatric and neurodevelopmental disorders like SCZ and autism. Although DA dysfunction and developmental deficits play a central role in the occurrence of these disorders, the molecular mechanisms driving the DA neuron function have remained largely unknown. Previously, several mouse models and zebrafish models have been developed to study the effects of 16p11.2 CNVs on the central nervous system development and animal behavior[11,48–50]. These studies have shown that mice with heterozygous 16pdel had significantly increased midbrain region[51] and deficits in basal ganglia- and DA-regulated circuitry, including increased numbers of Drd2+ medium spiny neurons in the striatum and downregulation of DA signaling components (Darpp32 and Drd1) in deeper cortical layers[48]. The Drd2+ MSNs also displayed synaptic deficits with increased EPSCs, and 16pdel mice also had reduced Th expression in the meso-diencephalon, which implicated deficits in dopamine signaling system[48]. In addition, 16pdel and 16pdup mice displayed opposite behavioral phenotypes, and these behavioral deficits were more severe in 16pdel mice[51]. 16pdel mice had poor adaptation to change, hyperactivity, learning deficits, repetitive and restricted behavior, sleep problems, and deficits in movement control and hearing[48,51]. Although several behavioral deficits in mice can be correlated with patient phenotypes, often the cellular and functional disease phenotypes in these animal models have been contradictory and have not recapitulated effectively the patient phenotypes present in clinic[13].

We differentiated and enriched DA neurons from iPSCs with NCAM$^{high+}$ sorting that resulted in homogeneous DA neuron populations of the control, 16pdel, and 16pdup cells. The limitation of generating homogeneous neuronal cell populations is that we may have dismissed some developmental cellular deficits that may be relevant for ASD phenotypes. Such deficits could include proliferation of precursor cells, differential generation of various neuronal subclasses, e.g., number of excitatory vs. inhibitory neurons[52,53], and alterations in astroglial cell development that is present in some ASD phenotypes and occurs in unsorted iPSC-derived neural cell populations during long-term culturing in vitro[54]. However, production of enriched DA neuron populations enabled us to detect disease phenotypes in a specific neuronal population that has a key role in the development of the 16p11.2 CNV disorders.

Our study revealed opposite changes in gene expression profiles of 16pdel and 16pdup DA neurons, which highlight the cellular processes regulating some of the opposing clinical phenotypes observed in patients with these CNVs. The discovery of increased expression of the gene ontology categories associated with programmed cell death, apoptosis, and cellular stress in the 16pdup cells indicates that these cells may be more vulnerable to cellular stress and death compared to control cells or cells with 16pdel. In addition, we found decreased expression of genes regulating neurogenesis, axon development, and neuronal body formation in the 16pdup DA neurons. Reduced expression of genes regulating neuronal development aligns with the reduced expression of TUJ1- and NCAM-positive neurons that we detected in the 16pdup cell population during differentiation. Such deficits in neuronal differentiation may, in part, explain the clinical phenotype of microcephaly detected in the patients with 16pdup[4]. In contrast to these findings, we detected enhanced expression of such genes in the 16pdel DA neurons and discovered a larger soma size of 16pdel neurons compared to control neurons. These genetic and morphological abnormalities may, in part, explain the clinical phenotype of macrocephaly in patients with 16pdel[15]. This result is also in agreement with the previous study of 16p11.2 CNV iPSC-derived cortical neurons that showed increased soma

size of the cells with 16pdel[39]. Interestingly, we did not detect differences in the cell proliferation between control cells and cells with 16p11.2 CNVs at the early stages of the neural floor-plate progenitor cell development, which is consistent with the data from previous studies of cortical cells with 16p11.2 CNV[39]. Taken together, our results suggest that processes regulating cellular growth, cellular death, and neurogenesis are central for brain growth deficits detected in the patients with this CNV[4,15].

The dopamine pathway plays a central role in regulation of reward-sensing circuitry and learning in the central nervous system. Genetic polymorphisms of the DA receptor D2 have been described in ASD, implicating that deficits in the function of reward-sensing circuitry may be related to abnormal social interaction and repetitive behaviors[55,56]. In line with this finding, here we found downregulation of DRD2 expression in 16pdup and 16pdel DA neurons. Previous studies have also shown that deficits in the dopamine receptor expression are common in patients with SCZ and patients with drug addiction[25,57,58]. Reduced DA receptor function induces increased DA neuron activation and increased dopamine secretion in patients with SCZ[57,59–61]. DAT mutations and altered DA homeostasis and hyperlocomotion are also present in ASD[62,63]. This is in agreement with our finding of increased DA neuron network activity and increased bursting in 16pdel networks. Related to neurotransmitter transportation, increased expression of norepinephrine transporter SLC6A2 was detected in our 16pdel DA neurons. Previous studies have shown that SLC6A2 is involved in cocaine and amphetamine transport to cells[64,65]. GRIA4 was also upregulated in 16pdel neurons. GRIA4 produces glutamate-ionotropic receptor AMPA type subunit 4, and it affects the excitability of neurons by regulating synaptic development and plasticity[66,67]. AMPA receptor and GRIA4 have also been associated with neurodevelopmental disorders with and without seizures[68] and SCZ[69], and drug and alcohol abuse[70]. Thus, upregulation of GRIA4 may be involved in the increased excitability of 16pdel neurons, and could be an interesting target for future drug screening studies for 16p11.2 CNV disorders. Compared to 16pdel neurons, we detected decreased expression of these genes in 16pdup neurons, along with decreased expression of KCND2 and opioid receptor OPRM1. KCND2 contributes to neuronal plasticity and affects memory, learning, and cognitive flexibility[71–73], and it has been associated with the development of ASD[74], Fragile X syndrome[75], and epilepsy[76]. KCND2 regulates dendritic excitability via activation of synaptic NMDA receptors and controlling the expression levels of synaptic NR2B/NR2A[77]. This makes KCND2 an interesting target molecule for future pharmacological studies of 16p11.2 CNV phenotypes. OPRM1 regulates dopamine secretion and reward circuit during drug or alcohol use, where drug binding releases excessive amounts of dopamine from the DA neurons and this positive-feedback loop contributes to the development of addiction[38]. Deficits in Oprm1 expression in knockout mouse models have also shown reduced responses to reward feeling and social cues, and autistic-like communication and motor deficits, and altered expression of genes associated with ASD in the striatum and dopamine pathway[78,79]. In line with these reports, we detected that the DA neurons with 16pdup displayed reduced number of spontaneously active neurons compared to control cell population and 16pdel neurons, and they had reduced excitability and decreased synaptic marker expression and neurotransmitter expression compared to 16pdel neurons. Thus, our results indicate that dopamine signaling and genes regulating drug use are altered in DA neurons with 16p11.2 CNVs, and these changes cause opposite functional phenotypes in deletion or duplication neurons and can be associated with reward feeling and behavioral deficits in autism and drug addiction.

For modeling of ASD with iPSC-derived neurons, previous studies of monogenic loss of *TSC2* have revealed morphological and functional phenotypes with the evidence of increased soma size, reduced synaptic function and hypoexcitability in TSC-deficient GABAergic inhibitory Purkinje neurons[80], and in TSC-deficient mixed cultures of cortical excitatory and inhibitory neurons[81]. For modeling SCZ, a previous study of iPSC-derived neurons described normal excitability and calcium transients in SCZ patient neurons compared to control neurons, although decreased numbers of neurites and synapses were detected in the SCZ neurons[82]. However, these functional phenotypes of the SCZ cultures were inconclusive since the studied pluripotent stem-cell-derived cell cultures were a mixture of glutamatergic and GABAergic neurons with a small fraction of DA neurons[82]. In studies of 16p11.2 CNVs, previous work on human iPSC-derived excitatory cortical neurons demonstrated that the 16pdel cells exhibited hypoexcitability due to increased soma size and cell volume rather than cell-intrinsic ion channel properties, whereas the 16pdup neurons had reduced soma size and dendrite length and did not have significant deficits on neuronal excitability[39]. In line with these morphological findings, here we also discovered larger soma size in 16pdel DA neurons compared to control cells or cells with 16pdup. However, the 16pdel DA neurons demonstrated increased excitability compared to control neurons, which is in contrast to the hypoexcitable neuronal phenotypes of the previous studies. One explanation for this contradiction is that DA neurons can alter properties of different voltage-dependent ion channels differently than glutamatergic or GABAergic neurons, and they can have both inhibitory and excitatory signaling features[83]. In addition, expression of synaptic markers SYN1 and PSD95 and synaptophysin were increased in the 16pdel DA neurons compared to control cells or cells with 16pdup. Reflecting the increased number of synapses, we also discovered that the 16pdel DA neurons displayed an increased frequency of the EPSCs compared to control neurons, which also differs from the cortical excitatory neuron phenotypes[39,81]. Thus, our patch-clamping experiments revealed a unique pathophysiology of the DA neurons with the 16p11.2 CNVs, which has not been previously detected in other neuronal subtypes associated with ASD or SCZ.

The 16pdel DA neurons displayed increased network activity, synchronization, and bursting phenotypes during network development compared to control cells. Interestingly, these data are in contrast to previous studies that detected reduced spontaneous network activity in the monogenic *KCTD13*-deletion iPSC-derived cortical neurons[14]. One explanation for these divergent findings is that DA neuron networks are functionally different from the excitatory cortical neuron networks since DA neurons can be self-regulated by the DA release and uptake via their dopamine autoreceptors[61,84,85]. Increased synchronous DA neuron activity and bursting are associated with increased DA release from the neurons[86]. In patients with OCD and in patients with repetitive behaviors, DA neuron dopamine secretion is abnormally increased and the expression of DA receptors is decreased leading to impaired learning and reward feeling in these patients[87]. In SCZ patients, hyperfunctional DA neurons have been detected in addition to reduced DA receptor expression, reduced DA sensitivity, and transportation in the brain[61]. Thus, our discovery of the hyperactivity and increased bursting of the 16pdel DA neuron networks indicated that these cells display similar disease phenotypes that are strongly associated with the development of OCD, SCZ, and repetitive behaviors in autism.

Our aim in this study was also to find a molecular pathway that could be pharmacologically manipulated to rescue the detected disease phenotypes in vitro. Previous studies have indicated the importance of normal *KCTD13* expression in the development

and functionality of cortical neurons. In iPSC-derived cortical neurons, *KCTD13* deficiency reduced the network activity, although RHOA expression was unchanged in the neurons[14]. In mouse models, loss of *Kctd13* reduced synaptic function of hippocampal neurons, which was associated with increased RhoA expression, but no alterations were detected in the brain size[13]. In contrast, other studies describe increased volumes of forebrain and midbrain regions in the *Kctd13*-deficient zebrafish and mouse models during development[12]. Further conflicting results were found in the functional development of mouse models with monogenic loss of *Kctd13*, which showed short-term memory deficits, RhoA-independent deficits in the spine maturation, and normal volumes of different brain regions[88]. In addition, the RhoA levels were unchanged in the hippocampus and cortex of the 16pdel mice compared to control mice, whereas RhoA levels were decreased in the 16pdup mice[88]. These contradictory results raise question about the role of *KCTD13* deficiency and regulation in the neurons. Thus, here we wanted to study in more detail the effects of KCTD13 and its downstream targets in our human DA neuron network model of 16p11.2 CNV. We detected reduced expression of KCTD13 in the 16pdel neurons as expected. Since KCTD13 regulates Cullin3–ubiquitin complex[89], the decreased activation of this ubiquitin–proteasome system can lead to increased expression of the RHOA in our DA neurons with 16pdel. Interestingly, we also detected significantly decreased expression of *EPHA7* in the 16pdel DA neurons compared to control cells. Previously, it has been shown that increased EPH signaling resulted in reduced RHOA activation via ubiquitination, and this phenomenon rescued excitatory synapse formation in the cell culture model of neurodevelopmental disorders[90]. Based on this previous study and our data, it is possible that decreased *EPHA7* expression in combination with reduced expression of KCTD13 detected in the 16pdel neurons contributes to increased RHOA expression in the DA neurons. Importantly, we discovered that pharmacological inhibition of the RHOA activation with Rhosin rescued the increased network hyperactivity and bursting activity in the 16pdel DA neurons. In line with our findings, previous studies showed that RHOA inhibition with Rhosin treatment rescued social stress-induced hyperexcitability in medium spiny neurons in a mouse model of anxiety[91]. No adverse effects were detected in stressed or non-stressed animals when treated with Rhosin for several days[91]. Rhosin enhances neurite outgrowth and also reduces cell growth and endothelial cell invasion via blocking RhoA activity, without affecting Cdc42 or Rac1 signaling[92]. This pathway specificity raises the possibility of Rhosin as a potential candidate for the treatment of neuropsychiatric and neurodevelopmental disorders. However, to date, no long-term safety studies with Rhosin have been conducted in animal models with neurological deficits. In conclusion, our study demonstrates that the RHOA pathway plays an important role in the development of the network hyperactivity in the human DA neuron networks with 16pdel, and inhibition of this pathway may be a potential target for developing personalized therapies for neuropsychiatric and neurodevelopmental disorders associated with CNV of this genomic region.

## Methods

**Human iPSCs lines with 16p11.2 CNVs**. Human iPSCs were derived from the fibroblast line GM08330, purchased from Coriell Institute for Medical Research and previously described[30]. CRISPR-Cas9-edited 16pll.2 deletion and 16pll.2 duplication iPSC lines were generated from the GM08330 iPSCs and characterized with RNAseq, aCGH, and western blot (WB) by Tai et al.[30], and provided to this study by Dr. Tai and Dr. Gusella from Massachusetts General Hospital (Boston, MA, USA). The Human Subjects ethics committee Boston Children's Hospital Institutional Review Board (IRB) approved the protocol (X09-08-0442) to study human iPSC lines at the Boston Children's Hospital (Boston MA, USA). The iPSC line karyotypes were confirmed by G-band karyotyping (Supplementary Fig. 1) and

SNP array (Supplementary Table 1a) (WiCell Cytogenetic Core)[93]. Three clones per genotype were used for differentiation and phenotypic characterization, nine iPSC lines in total (control; GM08330ctr, Cas9ctr clone1, Cas9ctr clone2, 16pdel; A3del, A5del, B8del, 16pdup; B9dup, C5dup, D9dup). The iPSC colonies were maintained on mTESR1 medium (Stem Cell Technologies) on Geltrex- (Thermo Fisher Scientific) coated plates and passaged every 5–7 days, and the media were changed every day. Cells were tested routinely to be mycoplasma-negative. The cells were analyzed with qRT-PCR to confirm pluripotency of the iPSCs (OCT4 and NANOG, Fig. 1c), and the expression of pluripotency markers OCT4, NANOG, and TRA1-60 were detected with ICC (Supplementary Fig. 1).

**DA neuron differentiation of iPSCs**. Human iPSCs were differentiated into DA neurons with the protocol that we had previously developed[31], with minor modifications outlined beneath. Three clones per genotype were used for differentiation experiments, nine iPSC lines in total (control; GM08330ctr, Cas9ctr clone1, Cas9ctr clone2, 16pdel; A3del, A5del, B8del, 16pdup; B9dup, C5dup, D9dup). Details of the number of replicates used are written after each experiment. In brief, the iPSC colonies were dissociated with Accutase (Innovative Cell Technologies) for 20 min at 37 °C and plated at high-density onto Geltrex- (Life Technologies) coated dishes with N2/B27-neural induction medium (Life Technologies) supplemented with Noggin 200 ng/ml and SB-43152 10 μM. On day 1, SAG 1 μM (Stemgent)[31] was added to the media. On day 3 of differentiation, CHIR-99021 3 μM (Stemgent) was added to the media to induce WNT signaling activation. Developing neural cells were cultured on high density to enhance the growth-factor production and exchange between cells. The basal neural induction media (DMEM/F12 + N2 supplement, Thermo Fisher Scientific) were gradually changed to neuronal differentiation base media (Neurobasal media + B27-AA supplement, Thermo Fisher Scientific), starting at day 4: 75/25, day 5: 50/50, day 6: 25/75, day 7: 5/95, and day 8: 0/100. On day 6, the neural precursors were passaged and re-plated into poly-D-lysine (PDL)- (50 μg/ml) and laminin- (10 μg/ml) coated plates in a media mixture of DMEM/F12/Neurobasal media with N2 supplement, B27 supplement, and growth factors Noggin 200 ng/ml (Peprotech Inc.), SB-43152 10 μM, SAG 1 μM (Stemgent), CHIR-99021 3 μM (Stemgent), and ROCK inhibitor (Rho kinase inhibitor, Y-27632, Cayman Chemical) 10 μM. FGF8b 100 ng/ml (Peprotech Inc.) was added to the cells on days 9–11 of differentiation, and from day 13 onwards, the cells were cultured on neuronal differentiation medium with factors: brain-derived neurotrophic factor (BDNF) 20 ng/ml, glial cell line-derived neurotrophic factor (GDNF) 20 ng/ml (Peprotech), DAPT (γ-secretase inhibitor) 2.5 μM (Cayman Chemical), cAMP 500 mM (Sigma), TGFb 1 ng/ml (Peprotech), and ascorbic acid 200 μM (AA, Sigma).

**Cell sorting with FACS**. After 27 to 32 days of differentiation, the iPSC-derived DA precursors were sorted with NCAM^high+ selection using FACS-Aria II and FACSDiva software v.8.0.3 (BD Biosciences) and low sheath pressure 20 PSI, and with sterile phosphate-buffered saline (PBS) as fluidics, and 100 μm nozzle[31]. Briefly, cells were dissociated into single cells with Accutase (Innovative Cell Technologies) and DNAse 100 U/ml 20 min at +37 °C. Cells were then washed and stained with NCAM antibody (anti-NCAM(CD56)-APC, SPM128, Novus Biologicals, NBP2-34397APC) 1:100 dilution to 500 μl of cell suspension for 20 min at room temperature. Cells were washed twice and suspended into 2% bovine serum albumin-Hank's balanced salt solution-penicillin/streptomycin (BSA-HBSS-P/S) and filtered through nylon mesh (35 μM, BD Falcon Cell Strainer cap). Sorted cells were collected into the Neurobasal medium supplemented with B27 and 1% BSA (Lampire Biological Products), centrifuged, and suspended on Neurobasal media with B27 (Thermo Fisher Scientific), BDNF 20 ng/ml, GDNF 20 ng/ml (Peprotech), DAPT 2.5 μM (Cayman Chemical), cAMP 500 mM (Sigma), TGFb 1 ng/ml (Peprotech), and AA 200 μM (Sigma). Cells were plated to PDL/laminin-coated coverslips for further differentiation, and samples for RNAseq and immunocytochemical analyses were collected after ~4 weeks of subculturing. At this time point, cells were also analyzed with patch-clamping method (days 60–65 of differentiation). In addition, sorted cells were also plated on HD-MEA chips (MaxWell Biosystems) or LD-MEA plates (Axion Biosystems) for functional analyses of the DA neuron network development (see below for protocols in detail). Three clones per genotype were used for FACS-sorting experiments, in total nine iPSC lines (control; GM08330ctr, Cas9ctr clone1, Cas9ctr clone2, 16pdel; A3del, A5del, B8del, 16pdup; B9dup, C5dup, D9dup). FACS sorting were done from clones; GM8330ctr n = 40, Cas9ctr clone1 n = 25, Cas9ctr clone2 n = 25, A3del n = 29, A5del n = 40, B8del n = 8, C5dup n = 35, D9dup n = 20, B9dup n = 12. FlowJo (v.9.9.6, BD Biosciences) was used to analyze the FACS data.

**RNA isolation, cDNA synthesis, and qRT-PCR**. The RNA isolation from iPSCs, NPCs, and iPSC-derived DA neurons was completed using RNeasy Plus Mini Kit (Qiagen) according to the manufacturer's protocol. The concentration and purity of the RNA samples were quantified using a NanoDrop spectrophotometer. Complementary DNA (cDNA) synthesis was performed using High Capacity cDNA Reverse Transcription Kit (Applied Biosystems) with 150 ng of the RNA per reaction. The qRT-PCR protocol was performed with SYBR Green 2× Master Mix (Applied Biosystems), with forward and reverse primers designed for the target gene sequences, and with purified water (Millipore) on MicroAmp Endura Plate

Optical 96-Well plates (Applied Biosystems). Primers were designed with Primer3 software (v.0.4.0) and purchased from Integrated DNA Technologies. The efficiency of the primers was verified with melting-curve analyses, and qPCR reaction was conducted with QuantStudio 3 software v.1.5.1 (Applied Biosystems). Used primers and their sequences are presented in Supplementary Table 2. For the analyses of the pluripotency and differentiation markers, two clones per genotype were used for the qRT-PCR analyses with three technical replicates per clone (Fig. 1c). For verification of the identified genes from the RNAseq analyses, two clones per genotype were used with biologically independent samples n = 3 control, n = 5 16pdup, and n = 3 16pdel, also two to three technical replicates per sample were ran per qPCR analyses (Fig. 2g–j and Supplementary Fig. 3i). For verification of the genes in the 16p11.2 gene region in iPSCs and NPCs, three clones per genotype were analyzed (n = 3 per group), and three technical replicates per each clone (control; GM08330ctr, Cas9ctr clone1, Cas9ctr clone2, 16pdel; A3del, A5del, B8del, 16pdup; B9dup, C5dup, D9dup, Supplementary Fig. 1c, d).

**Transcriptional gene expression analyses**. At a total differentiation time of ~50 days, we collected the FACS-sorted DA neurons into RNA-lysis RLT-Plus buffer + β-mercaptoethanol (Qiagen). For isolation and purification of the RNA, we used RNeasy Mini Plus Kit according to the manufacturer's instructions (Qiagen). Only high-quality samples with RNA integrity values 8.5–10 were used for cDNA-library preparation. Purified total RNA (500 ng/sample) was used for cDNA-library preparation using Illumina TruSeq Stranded mRNA Sample Preparation Kits. The dsDNA libraries were quantified by Qubit fluorometer, Agilent TapeStation 2200, and RT-qPCR using the Kapa Biosystems Library Quantification Kit. The sequencing was done with Illumina NextSeq500 run with single-end 75 bp reads (Illumina sequencer, Illumina Inc.) at the Molecular Biology Core Facilities at the Dana-Farber Cancer Institute (Boston). Two clones per each genotype were used for RNAseq (control; GM08330ctr, Cas9ctr clone1, 16pdel; A3del, A5del, 16pdup; C5dup, D9dup). The number of biological replicates per genotype were control n = 4, 16pdel n = 7, and 16pdup n = 6 (Fig. 2). The number of biological replicates per clone were GM08330ctr n = 1, Cas9ctr clone1 n = 3, A3del n = 4, A5del n = 3, C5dup n = 3, and D9dup n = 3 (Fig. 2 and Supplementary Fig. 3g).

We performed the data analyses as previously described[80]. Briefly, initial quality control for the sequencing data with fastQC (v.0.11.5). Reads were then mapped with STAR (v.2.7.0) using an index created from release 91 of the HG38 build of the genome from Ensembl with associated transcript annotations. Cufflinks (v.2.2.1) was used to create a merged transcriptome assembly from all samples, and this reference was used to quantify gene abundance from all samples. We performed differential expression analysis with LIMMA (v.3.46.0). We did Weighted Gene Co-expression Network Analysis using the WGCNA package (v.1.70-3) in R (v.4.0.3). We used the top 10,000 most highly expressed genes for construction of the gene network. After that, we analyzed the adjacency matrix for these genes with a beta = 6, and this adjacency matrix was used to calculate the topological overlap matrix. Hierarchical clustering of the topological overlap matrix was done using the "average" algorithm, and modules were identified using dynamic tree cutting. The first-principal component was calculated using the module eigengene function. To identify co-expression modules that were related to copy number of the 16p11.2 gene region, we performed a Pearson's correlation between copy number (deletion—1, control—2, and duplication—3) and the module eigengenes. The M19 and M25 modules had the greatest positive and negative correlations. Gene ontology analysis was performed using DAVID v.6.8 (https://david.ncifcrf.gov/). Ensembl identifiers corresponding to genes in either the M19 or M25 module were used as input into DAVID. Gene ontology categories were selected based on specificity and nonredundancy. IPA (version 57662101) was used for pathway enrichment analysis in Ingenuity Knowledge Base (www.qiagen.com/ingenuity). Genes with a kME > 0.5 in either the M19 or the M25 co-expression modules were included in the analysis (Supplementary Fig. 4).

**Immunocytochemistry**. Cells were analyzed with immunocytochemical staining to confirm patterning of floor-plate progenitors with FOXA2 antibody at day 5 of differentiation, and TUJ1 and TUJ1/FOXA2 antibody for neural cells and Ki67 for proliferative cells at days 5 (n = 3 per genotype; number of clones: GM control n = 3, Cas9ctr clone2 n = 3, B8del n = 3, B9dup n = 3) and 17 (n = 6 per genotype; number of clones: GM control n = 3, Cas9ctr clone2 n = 3, A3del n = 3, A5del n = 3, C5dup n = 3, D9dup n = 3). Cells were analyzed at day 60 of differentiation for TH/TUJ1 expression (n = 6 control, n = 4 16pdel, n = 3 16pdup; number of clones: GM control n = 4, Cas9ctr clone2 n = 2, A3del n = 2, B8del n = 2, B9dup n = 3), TH/NURR1 (n = 3–4 per genotype, GM control n = 4, A3del n = 4, B9dup n = 3), TUJ1 (n = 6 control, n = 3 16pdup, n = 5 16pdel, number of clones: GM control n = 4, Cas9ctr clone1 n = 2, A3del n = 3, A5del n = 1, B8del n = 1, C5dup n = 1, B9dup n = 2), SOX2/Nestin (n = 4 control, n = 3 16pdup, n = 3 16pdel; number of clones: GM control n = 1, Cas9 clone 2 ctr n = 3, A5del n = 1, B8del n = 2, C5dup n = 3), and GABA (n = 7 per genotype; number of clones: GM control n = 1, Cas9ctr clone2 n = 6, A3del n = 1, B8del n = 6, B9dup n = 1, C5dup n = 6). Synaptic markers SYN1 and PSD95 were characterized with co-staining of TUJ1-positive neurites (n = 5 per genotype; number of clones: GM control n = 2, Cas9ctr clone1 n = 3, A3del n = 3, B8del n = 2, C5dup n = 3, B9dup n = 2), and SYP1 (n = 4 per genotype; number of clones: GM control n = 2, Cas9ctr clone1 n = 2, A3del n = 2, A5del n = 2, C5dup n = 2, D9dup n = 2) and DAT expression with

co-staining of TH-positive neurites from the patch-clamped cells. For immuno-cytochemical characterization the cells were first fixed with 4% paraformaldehyde for 20 min at room temperature. After washing with Dulbecco's PBS, nonspecific staining was blocked with 5% BSA–PBS 0.1% Triton X-100, and primary antibodies were diluted into 1% BSA–PBS–0.1% Triton X-100 and incubated with the cells overnight at +4 °C. Table of used antibodies is in Supplementary Table 3. After dPBS washes, cells were incubated with secondary antibodies (Supplementary Table 4) for 1 h at room temperature. Nuclei were stained with Hoechst 33258 (Thermo Fisher Scientific) for 5 min at room temperature.

**High-content imaging and analyses.** We imaged the stained coverslips (3–6 coverslips/genotype) and 96-well plates (8–12 wells/genotype) with ImageXpress MicroXLS Widefield High-Content Molecular Device microscope (Molecular Device, LLC, MetaXpress v.6.6.2.46). We used ×20 magnification, and z-stack with seven steps and 2 μm thickness/step, stacked images were combined with maximum resolution projection. We used Molecular Devices MetaXpress software (v.6.5.2.351) to design an algorithm for identification of selected features that were quantified from each image (Figs. 1 and 3 Supplementary Fig. 3). Representative images of the IXM algorithm masks used for TH/TUJ1 quantification are shown in Supplementary Fig. 6.

We used ImageJ/Fiji (v.1.0) software for TH-positive soma area analyses, and three clones per genotype were quantified, 16pdel; A3del, A5del, B8del, and 16pdup; B9dup, C5dup, D9dup, and for control: GM08330 and Cas9 clone1, total $n = 6$ for vehicle and $n = 6$ for Rhosin treatments per genotype (Fig. 8 and Supplementary Fig. 8). The number of cells analyzed per clone were GM08330ctr + veh $n = 192$ (batch1), $n = 121$ (batch2), GM08330ctr + Rhosin $n = 206$ (batch1), $n = 104$ (batch2), Cas9ctr clone1 + veh $n = 128$, Cas9ctr + Rhosin $n = 135$, A3del + veh $n = 157$, A3del + Rhosin $n = 107$, A5del + veh $n = 230$, A5del + Rhosin $n = 268$, B8del + veh $n = 146$, B8del + Rhosin $n = 144$, B9dup + veh $n = 139$, B9dup + Rhosin $n = 154$, C5dup + veh $n = 130$, C5dup + Rhosin $n = 153$, D9dup + veh $n = 221$, and D9dup + Rhosin $n = 241$.

**Confocal imaging.** We imaged the stained coverslips with LSM700 laser scanning confocal microscope with ×63/1.40 Oil DIC M27 (Carl Zeiss LSM700, German) with Zen-Blue software v.8.1. (Zeiss), and maximum projections of the images were created from the z-stacked images.

**Western blot.** Cells were lysed to RIPA-lysis buffer (Santa Cruz Biotechnology Inc.) supplemented with protease inhibitors, sodium orthovanadate, phenylmethylsulfonyl fluoride (Santa Cruz Biotechnology Inc.), and phosphatase inhibitor cocktails (Bimake). Protein concentrations were analyzed with BCA Kit (Thermo Fisher Scientific), and 15 μg of protein was loaded/well into 4–20% Criterion TGX gels (Bio-Rad). Gels were blotted into polyvinylidene difluoride membranes and total proteins were analyzed with Licor Revert Total protein Stain Kit (LI-COR Biosciences). Blots were blocked with Intercept-TBS blocking buffer (LI-COR Biosciences). Primary antibodies (Supplementary Table 3) were diluted into TBS-T with 1% BSA, and incubated on the blots overnight. Blots were washed three times with TBS-T, and secondary antibodies were diluted into TBS-T 1% BSA and incubated on the blots for 1 h at room temperature (Supplementary Table 4). Blots were detected with Licor-CLX (LI-COR Biosciences). Two control lines and three clones per 16pdel and 16pdup lines were analyzed (control; GM08330ctr, Cas9ctr clone1, 16pdel; A3del, A5del, B8del, 16pdup; B9dup, C5dup, D9dup). The number of replicates per genotype with vehicle/Rhosin treatment were CULLIN3 $n = 6$, RHOA $n = 7$, pCofilin/Cofilin $n = 10$, and LDHA $n = 9$ (Fig. 8 and Supplementary Fig. 8) and without treatments KCTD13 $n = 8$, RHOA $n = 8$ (Fig. 7). Uncropped WBs are in the Source data file.

**Patch clamping of the iPSC-derived DA neurons.** Approximately after 60–65 days of total differentiation time, the iPSC-derived DA neurons were characterized with patch-clamp recordings with method previously described in ref. [94]. Briefly, DA neurons were visualized using an upright Olympus BX51 microscope with epifluorescence capabilities. Current- and voltage-clamp data were recorded and analyzed with a SutterPatch IPA (v.2.1.0. Sutter Instruments, Sunnyvale, CA). Glass pipettes were formed from borosilicate glass (3–5 MΩ) on a Sutter Puller (P-97 Sutter Instrument) filled with intracellular solution: 130 mM K-Gluc, 10 mM K-HEPES, 4 mM KCl, 10 mM Na-phosphatase, 4 mM Mg-ATP, 2 mM Na-ATP, 0.3 mM Na-GTP MgCl₂, OsM 295 mOs pH 7.33 with KOH. Cells were visualized with Alexa Fluor 488 nm dye (1 μM, Thermo Fisher). Extracellular recording solution, artificial cerebral spinal fluid (ACSF), contained the following: 125 mM NaCl, 3 mM KCl, 2.5 mM CaCl₂, 1.3 mM MgSO₄, 1.25 mM NaH₂PO₄, 26 mM NaHCO₃, 13 mM glucose pH to 7.4 with NaOH, 300 mOsMo. ACSF solution was continuously oxygenated during experiments with 95% O₂ and 5% CO₂ to maintain pH 7.4. Three iPSC clone lines per genotype were recorded with patch-clamp (control; GM08330ctr, Cas9ctr clone1, Cas9ctr clone2, 16pdel; A3del, A5del, B8del, 16pdup; B9dup, C5dup, D9dup). The number of cells patched per genotype were control $n = 26$, 16pdup $n = 23$, and 16pdel $n = 23$ (Fig. 4). The number of cells patched per clone were GM08330ctr $n = 6$, Cas9ctr clone1 $n = 16$, Cas9ctr clone2 $n = 9$, A3del $n = 5$, A5del $n = 8$, B8del $n = 9$, C5dup $n = 6$, D9dup $n = 6$, and B9dup $n = 11$.

**Generation of excitability curve and quantification of Rheobase.** Cells were injected with 500 ms of depolarizing currents of increasing amplitude while being held at I-Hold = −60 mV. Rheobase, a value that describes the minimum current need to evoke an AP, was generated using SutterPatch/IgorPro8 software v2.1.0. We averaged input–output excitability curves for all neurons within a given condition (IgorPro8 Wave_Average command) and generated an average excitability curve per condition. The point where the average excitability curve crossed the y-axis equals to 1 AP is presented as the Rheobase.

**HD-MEA recordings of the iPSC-derived DA neurons.** For functional analyses of the developing iPSC-derived DA neuron networks, we used Maxwell Biosystems CMOS-HD-MEA system (MaxOne System, MaxWell Biosystems AG, Switzerland)[40,41]. MaxOne chip contains 26,400 platinum electrodes in a sensing area of $3.85 \times 2.10$ mm² with 17.5 μm center-to-center pitch, 3265 electrodes/mm² density, and 1024 configurable low noise readout channels ($2.4 \mu V_{r.m.s.}$ in the 300 Hz–10 kHz band) with a sampling rate of 20 kHz/s at 10-bit resolution.

HD-MEA-chips were pre-coated with polyethylenimine (PEI) 0.1% (Sigma-Aldrich) in borate buffer (Thermo Fisher Scientific) overnight at room temperature and washed three times with dPBS, and then we added laminin 20 μg/ml in DMEM/F12 for coating of the chips at +37 °C overnight. Prior to plating the iPSC-derived DA neurons on the chips, the cells were sorted with FACS-Aria II (BD) with NCAM^high+ selection. We plated 125,000 of sorted neurons onto the pre-coated chips in DA neuron media with ROCK inhibitor 10 μM and BSA 0.25% and growth factors: BDNF 20 ng/ml, GDNF 20 ng/ml (Peprotech), DAPT 2.5 μM (Cayman Chemicals), cAMP 500 mM (Sigma), TGFb 1 ng/ml (Peprotech), and AA 200 μM (Sigma). The next day, 8000 human astrocytes (Ncardia) were added on top of the neurons. After 12 days of plating the cells on the chips, recordings were started, and the activity of the developing networks was measured over 4 weeks (Figs. 5 and 6 and Supplementary Fig. 5a).

For the recordings, we used MaxLab Live Software (v.20.1.6. MaxWell Biosystems AG, Switzerland). Briefly, spontaneous activity of the neurons was measured using the Activity Scan Assay where the whole chip area was scanned with a sparse recording (30 s/configuration, seven configurations). The active neurons were automatically identified, based on the firing rate and spike amplitude obtained from the Activity Scan. Based on the activity of the neurons, the most active sensors were routed for the creation of the network configuration of the 1024 most active sensors. Selected active sensors were then simultaneously recorded using the Network Assay to investigate the evolution of the network dynamics through the extracted burst features (recording time 10 min/configuration, 1 configuration). For the whole chip scan, we used our in-house made configuration of $23 \times 22$ areas of a total of 1024 sensor/area (recording time 2 min/configuration, 25 configurations). HD-MEA recordings were done from three clones per genotype (control; GM08330ctr, Cas9ctr clone1, Cas9ctr clone2, 16pdel; A3del, A5del, B8del, 16pdup; B9dup, C5dup, D9dup). The number of chips per control were $n = 6$, 16pdup $n = 7$, and 16pdel $n = 4$, (Figs. 5 and 6). The number of clones were GM08330ctr $n = 3$, Cas9ctr clone1 $n = 1$, Cas9ctr clone2 $n = 2$, A3del $n = 2$, A5del $n = 1$, B8del $n = 1$, C5dup $n = 3$, D9dup $n = 2$, and B9dup $n = 2$ (Supplementary Fig. 7).

**Preprocessing of the HD-MEA data.** During the data processing, we discarded the values of invalid sensors (saturated sensors). For spike detection, we used bandpass filtering of data with a butterfly filter of 300–7000 Hz. We calculated median absolute deviation (MAD) per sensor and every negative peak <−8 MAD was considered a spike. We considered a sensor active if the detected spike rate was >0.05 Hz. For analyses of the frequency of the active sensors, we calculated the number of spikes detected per second. For the analyses of the synchronicity, we binned the data at 1 ms and calculated the cross-correlation between all active sensors at zero lag (this indicates which sensors regularly spike within 1 ms of one another). All pairs of active sensors that were cross-correlated above threshold 0.25 were considered "synchronized pairs." If a sensor is a member of at least one synchronized pair, it was considered synchronized. This analysis was conducted on data from whole chip scans, i.e., from 25 recordings of subarrays of the whole array. This means the synchronicity was always calculated locally, and our results do not capture the possible synchronicity between distant sensors across the array.

**Burst analyses on the HD-MEA data.** Burst detection was done according to previously published algorithm by Kapucu et al.[43]. The minimal number of spikes in a burst was set to 3, other parameters were kept the same as in the original algorithm. The analyzed datasets were from the results of the whole-array recordings from 25 different subarrays. The burst characteristics were calculated from the distributions given by all the bursts detected on all the sensors combined over all subarray recordings; e.g., the average burst duration in a recording is the average over all the detected bursts in the 26,400 sensors taken together. The only timepoints that were taken into account in the burst analyses had at least 100 bursts detected over all combined sensors. Burst duration is defined as the time between first and last spike within the burst. Time between burst interval was defined as the interval between first spikes of consequent bursts. Time within burst interval was defined as the interval between spikes within a burst. The number of spikes per burst was calculated as well.

A raster plot (in general) visualizes the detected spike patterns for a selected set of sensors: the *x*-axis represents time and the *y*-axis represents the sensors. That is, each dot is a spike detected at a certain time (*x*-axis) in a certain sensor (*y*-axis).

To visualize the analyzed burst data, we generated raster plots for the sensors with the highest burst rate and sensors with most spikes per bursts. For this analysis, we started from the bursts detected in each sensor across the array[43]. For the visualization of burst rate, the average number of highest burst rate per sensor was calculated, and the sensors were ranked based on this result: from sensors with the highest burst rate to lowest burst rate. Then, we selected the top 200 sensors to make representative raster plots of the data (*y*-axis 0 is the sensor with the highest average burst rate, Fig. 6e). For plotting of the sensors with most spikes per burst, the average number of spikes per burst per sensor was calculated per sensor, and the sensors were ranked based on this result: from sensors with highest average number of spikes per burst to lowest average number of spikes per burst (*y*-axis 0 is the sensor with the highest average number of spikes per burst, Fig. 6f). These raster plots are compiled from active sensors from different subarrays of each recording, and thus the spike traces shown for each selected sensor are therefore not necessarily recorded simultaneously.

**Low-density MEA recordings of the iPSC-derived DA neurons.** Multi-well LD-MEA platform (Axion BioSystems) was used for drug-treatment assay and for the verification of the network phenotypes on the 16p11.2 CNV DA neuron networks compared to control cells. CytoView-MEA plates with 48-wells/plate and 16 electrodes/well were pre-coated with PEI 0.1% in the borate buffer overnight at room temperature. Plates were washed three times with dPBS followed by laminin 20 μg/ml at +37 °C overnight. Cells were plated after FACS sorting as single cells 120,000 cells/well in a media: Neurobasal, B27, 0.5% BSA, BDNF 20 ng/ml, GDNF 20 ng/ml, AA 200 μM, cAMP 0.5 mM, TGFb 1 ng/ml, and DAPT 1.25 μM. The next day, human astrocytes (Ncardia) 8000 cells/well were plated on top of the neurons. Neuronal activity was recorded every other day following 3–4 weeks. We treated the neurons with Rhosin 1 μM on MEA after 5 days of plating on MEA up to 28 days, and the drug or vehicle was added into the media, and for the cells, every other day during the media change, and the plates were recorded 48 h after media change. LD-MEA data were analyzed as previously described in ref. 95. Briefly, for the recordings we used the Maestro MEA amplifier (M768-GLx, Axion BioSystems) that can record the spontaneous activity of the networks with each electrode of the grid/well. For each time point, the network activity was recorded for 5 min. The recorded data were sampled at 15 kHz, digitized, and analyzed using Axion Integrated Studio software (v.2.5.1. Axion BioSystems), and AxIS Metric Plotting Tool (v.1.5.11) with a 200 Hz high-pass filter and 2500 kHz low-pass filter and an adaptive spike detection threshold set at 5.5 times the standard deviation for each electrode with 1 s binning. Data were analyzed from all the wells that contained >3 active electrodes/well. LD-MEA recordings were done from control; Cas9ctr clone1, 16pdup; C5dup, 16pdel; A5del. The number of biological replicates per condition were control + veh *n* = 16, 16pdup + veh *n* = 16, 16pdel + veh *n* = 16, control + Rhosin *n* = 7, 16pdup + Rhosin *n* = 6, and 16pdel + Rhosin *n* = 6 (Fig. 7d–i and Supplementary Fig. 5b).

**Statistical analyses.** Statistical analyses were performed with GraphPad Prism version 8.2.1. First, we performed a normality test for all the datasets to determine Gaussian distribution. Based on this test, we performed parametric or nonparametric tests between groups: two-way analysis of variance followed by Tukey's multiple comparisons test or Kruskal–Wallis test followed by Dunn's multiple comparisons test, and *p* values were adjusted for multiple comparisons. Unpaired *t* tests (parametric and nonparametric) with one- and two-tailed *p* values were performed in cases where comparisons were independent of each other. *P* values are presented as follows: $*p < 0.05$, $**p < 0.01$, $***p < 0.001$, and $****p < 0.0001$. Figure legends include the details of the statistical methods used for analyses of each dataset.

**Reporting summary.** Further information on research design is available in the Nature Research Reporting Summary linked to this article.

## Data availability

The transcriptional RNAseq data have been deposited to the European Genome-phenome Archive (EGA) under the study accession number: EGAS00001005137, and the data accession number: EGAD00001007072. All other data are available within this manuscript and its Supplementary information. Additional information is available from the corresponding author upon reasonable request. Source data are provided with this paper.

## Code availability

Accompanying code for processing and analyses of the HD-MEA-data is available at the webpage https://github.com/HannahPinson/HD-MEA-data-analysis.

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

## Acknowledgements
We would like to thank the Flow Cytometry Core at the Hematology Oncology Division at Boston Children's Hospital for assistance with the neural cell sorting and Dr. Lee Barrett for assistance with the image analyses at the Human Neuron Core at Boston Children's Hospital (BCH IDDRC, U54HD090255). We are grateful to Drs. Ville Kujala, Meera Modi, Denise McGinnis, and members of the Sahin lab for critical reviewing of the manuscript and helpful comments. This study was supported by Tommy Fuss Center for Neuropsychiatric Disease Research and the Rothberg Family Fund for Cognitive Science at Boston Children's Hospital and MIT. H.P. was supported by a fellowship from the Research Foundation Flanders (FWO-Vlaanderen) under Grant No. 11A6819N. R.S.S. was supported by F32NS100338 and K99NS112604. K.D.W. was supported by the American Academy of Neurology Neuroscience Research Training Scholarship and NIH NINDS 5K08NS112598. J.F.G., M.E.T., and D.J.C.T. received support from NIH Grants NS093200, HD096326, and GM061354. C.A.W. is an Investigator of the Howard Hughes Medical Institute. C.A.W. is supported by grants from the NINDS (R01NS032457) and NIMH (U01MH106883). M.S. is supported by NINDS R01NS113591.

## Author contributions
M.S. contributed to the design of the study, experimental planning, neuronal cell differentiation and characterization, cell sorting, pharmacological compound testing, HD-MEA-and LD-MEA recordings, data analyses, and writing of the manuscript. H.P. contributed to HD-MEA data analyses. R.S.S. performed the patch-clamp recordings and data analyses. K.D.W. performed the transcriptional gene expression analyses. P.V. contributed to qRT-PCR experiments and analyses. D.J.C.T. contributed to the generation of CRISPR-cas9 induced 16p11.2 CNV iPSC lines. J.F.G. contributed to experimental design and discussion of the experiments. M.E.T. contributed to experimental design and discussion of the experiments. C.A.W. contributed to experimental design and discussion of the experiments. M.T. contributed to experimental design and discussion of the experiments. M.S. contributed to the design of the study, experimental planning, discussion of the experiments, and writing of the manuscript. All authors read and edited the manuscript. H.P. and R.S.S. contributed equally to this work.

## Competing interests
The authors declare no competing interests.
