## [Peer Review File · Nature Communications]

Reviewers' Comments:

Reviewer #1:

Remarks to the Author:

Review NCOMMS-20-19298

This manuscript is well written, and for the most part, results are consistent with the interpretation provided by the authors. However, there are some aspects that needed to be addressed to strengthen the study.

1. Junction site sequencing of the iPSC lines with 16p11 deletion and duplication has been described in Ref 31. Did the CNV iPSC clones used in the present study exhibit different junction sequences (indels)? Will this differentially affect expression of genes located in the segmental duplications (cleavage sites) and ultimately the phenotype?
2. Karyotype analysis of iPSC by G-banding (Suppl. Figure 1) is of low resolution. Since genomic instability of iPSC is a serious confounder of iPSC studies, I would strongly suggest to confirm a normal karyotype by higher resolution array-based methods.
3. (Gene expression analyses) Which genes from the 16p11 region are already expressed at the iPSC stage or the NPC stage in the present study? Will differential expression affect pluripotency or early differentiation?
4. Did the authors perform pathway analysis based on their data from differential gene expression analysis (e.g. as in Ref. 14)? Did unbiased pathway analysis point to small GTPase (Rho) signaling (present study) or to MAPK/ERK signaling (shown in 16p11 CNV mouse models, e.g. Pucilowska, J Neurosci, 2015; Blizinsky, PNAS, 2016)? Are any of the functional categories they identified by weighted co-expression analysis and gene ontology analysis (e.g. cell death, oxidative stress, metabolic processes etc.) affected in the 16p11 CNV iPSC-derived dopaminergic neurons?
5. The authors analyzed differential expression of selected genes from the two modules and discuss their potential (patho)physiological function in detail. Might any of these deregulated genes (e.g. GRIA4, KCND2, or others identified by RNA-seq) contribute to increased excitability in 16p11 del neurons and represent a druggable therapeutic target? Are there any data regarding the potential side effects of long-term, systemic inhibition of RhoA signaling? Please discuss.
6. In Figure 3 A, C numerous Hoechst-positive nuclei are visible that do not show TUJ1/TH immunoreactivity. The authors should indicate the percentage of non-neuronal (TUJ1 negative) cells after NCAM sorting and maturation. Are there any hints from their RNA-seq analysis, which other cell types are detectable in these cultures? How did the authors ascertain that they record from dopaminergic neurons during patch-clamping (e.g. by transduction with a TH_GFP lentivirus)?
7. Figure 7 shows a clear trend towards increased RHOA protein expression in 16p11 dup iPSC-derived neurons compared to controls, however, this trend is not visible in the firing rate. Please discuss. The authors suggest that the decrease in KCTD13, which acts as an adaptor protein for Cullin-3 ubiquitin ligase, underlies the increase in the substrate RHOA. To make this more convincing, protein expression of Cullin-3 should be analyzed by immunoblotting as well.
8. (Discussion) Portmann et al (Ref. 57) and Horev et al (PNAS, 2011) analyzed alterations in dopamine-modulated brain circuits and midbrain dopaminergic neurons in 16p11 CNV mouse models. Their findings are highly relevant to the present study and should be discussed.
9. (Discussion) It should be mentioned that RhoA levels were unchanged in the cortex and hippocampus of 16p11 del mice and decreased in 16p11 dup mice in the study by Arbogast et al (Ref. 77).
10. Please provide a web link for your publicly available RNA-seq dataset.

Minor points

11. The authors performed NCAM sorting to enrich iPSC-derived neurons. Although cell sorting will decrease heterogeneity, it may also obscure ASD-relevant, early neurodevelopmental alterations induced by 16p11 deletion/duplication. This downside should be discussed.
12. In Figure 1H large aggregates of TH/TUJ1-positive iPSC-neurons are visible, which are notoriously difficult to quantify by HCI analysis. The authors should show a representative segmentation of this microscopic image by their HCI analysis software.

13. Spike rate and burst rate of wild-type iPSC-derived neurons on MEA usually increase during maturation. The authors should discuss, why this increase is not detected in the present study.
14. It might be helpful for many readers to provide a supplementary table listing disease-relevant findings in iPSC models / mouse models of the 16p11 del syndrome and in human 16p11 del carriers diagnosed with ASD (SCZ).

Reviewer #2:

Remarks to the Author:

The manuscript by Sundberg et al. describes the thorough characterisation of isogenic iPSC lines with deletions and duplications in the 16p11.2 locus, which is associated with the development of schizophrenia and autism. The study is well-performed, containing a thorough characterisation of the neurons through RNA sequencing, analysis of synapse formation, electrophysiology and MEA. The study shows altered electrophysiological features, and in particular altered burst firing patterns in the 16p11.2 CNV neurons. Many of the observations are in line with observations from a previous study (Deshpande et al., 2017), also studying 16pDel and 16pDup in human iPSCs. Nonetheless, the observations also provide some unique and relevant insights to the potential pathophysiology of neuropsychiatric disorders through RNA sequencing and MEA measurements. The study further shows that some of the observed abnormalities can be rescued by treatment with the Rho inhibitor Rhosin. The latter is however not novel, as it builds on a previous finding by Escamilla et al, published in Nature in 2017. Overall, the study contributes with solid and thorough validation in a human isogenic model system of previously presented hypotheses in the field, and it provides some new insights regarding RNA expression patterns, synchronous activity and burst firing patterns in the mutated cell lines. A few points require further attention:

- The study emphasises the relevance of studying the 16p11.2 CNVs in the context of midbrain DA neurons, which are known to play a key role in neuropsychiatric disorders. This is further a key feature of the current study to distinguish from the Deshpande et al., study which uses iPSCs. However, although the authors show that the progenitor cells in the cultures express FOXA2 at an early stage (day 5. Fig. 1C+E), there is not provided much evidence to support that the DA neurons in the cultures are truly of a midbrain DA phenotype. The FOXA2 expression in the cultures seems for some reason to be unstable, since a significant proportion of the cells are FOXA2-neg at 17 days of differentiation. Please provide stainings of FOXA2/LMX1A and of NURR1/TH cells at a late time point in the culture (i.e. day 60) + and quantifications of at least one of these stainings.

- It would be beneficial to show the differentiation protocol with addition of the different factors in the schematic in Fig., 1A

- From the methods section, it appears that "Three clones from each genotype were used for the DA neuron differentiation". It also appears that " All experiments were repeated at least three times with independent differentiation batches of neurons". It is however unclear if all three clones were used for all experiments, and how many biological replicates are included for each clone in each experiment. For some of the key experiments, the data should be shown independently for each clone in order to validate that all the clones from the same genotype are displaying the same phenotype. This is to ensure that it is not one or two clones in the dataset which are driving the differences. For each analysis it should be noted in the figure legend or materials and methods how many clones were used and how many biological replicates for each clone are included in the data.

- The manuscript is very lengthy and could benefit from significant trimming. In particular, there are several references to each individual observation in the results and discussion sections, which leaves an impression of a scattered manuscript with lack of focus.

- Line 143: "of these neurons >50-75% were FOXA2 positive". Should read either ">50%" or "50-75%"

Reviewer #3:

Remarks to the Author:

In the present manuscript, Sundberg M et al utilize dopaminergic neurons derived from human iPSCs carrying CNVs of 16p11.2 duplication and deletion induced by CRISPR-Cas9 gene editing to investigate network functions and alterations seen in neuropsychiatric/neurodevelopmental disorders as schizophrenia and autism. In this work, neuronal differentiation is well characterized, and cultures are enriched for neurons with a FACS step previous gene expression analysis, electrophysiological recordings and network investigations. Electrophysiological assessments are performed both with whole cell patch clamp recordings and, at network level, with MEA chips. The authors report that 16p11.2 deletion results in synaptic connectivity and excitability changes in dopaminergic neurons which lead to hyperactive networks in vitro. RhoA inhibition obtained with Rhosin administration rescued network hyperactivity.

Overall the work is sound, well designed, and of interest for the field.

In my opinion, few points should be addressed before considering this manuscript for publication:

-The Results section is lacking details about the Statistics used to analyze different experiments as well as N numbers. This information should be included.

-In Figure 1E-G, the authors report an increased number of Tuj1 positive cells in the deletion condition, compared to control and duplication. Same for the NCAM analysis in G. However, they report that the ratio between Tuj1 and TH positive neurons is maintained among the different conditions. Do the authors know the fate of these "other" neurons and to which extent are they present after FACS enrichment? The data seem to suggest that these neurons are not dopaminergic. This could potentially play a role in the network studies. This point should be clarified.

-Heatmaps in Figure C and D are not of clear reading and differences in gene expression are not always obvious. I suggest to use a lighter color for the 0 value instead of black.

-Figure 3C, differences in SYP1 densities in the three conditions are difficult to appreciate, I suggest using different microphotographs.

-Figure 4, since the authors report differences in soma area in the 16p11.2 deletion condition, did they see also differences in cell capacity from whole cell recordings? Also, a separate bar graph should be add with Rheobase quantifications.

-Figure 7, the functional data obtained after Rhosin administration should be strengthen with investigations on the morphological and molecular changes of the dopaminergic neurons within the network. Moreover, RhoA inhibition should be validated in the culture to exclude off target effects.

Point-by-point response to reviewer comments - NCOMMS-20-19298

Reviewer #1 (Remarks to the Author):

This manuscript is well written, and for the most part, results are consistent with the interpretation provided by the authors. However, there are some aspects that needed to be addressed to strengthen the study.

1. Junction site sequencing of the iPSC lines with 16p11 deletion and duplication has been described in Ref 31. Did the CNV iPSC clones used in the present study exhibit different junction sequences (indels)? Will this differentially affect expression of genes located in the segmental duplications (cleavage sites) and ultimately the phenotype?

The CNV iPSC clones used in this study were identical to the clones used in ref 31 (Tai et al., 2016, Nature Neuroscience). The SCORE guides were identical, we do not have any evidence of additional alterations, and none of the SCORE guides were within a segdup gene. Based on the qRT-PCR analysis, the expression levels of cleavage site genes *SLX1A*, *SULT1A4* and *BOLA2* are linear-correlated with copy number and no additional effects on these been detected in 16pdel and 16pdup neural precursor cells. Therefore, we don't anticipate there to be an effect on the cleavage sites or the phenotypes detected.

2. Karyotype analysis of iPSC by G-banding (Suppl. Figure 1) is of low resolution. Since genomic instability of iPSC is a serious confounder of iPSC studies, I would strongly suggest to confirm a normal karyotype by higher resolution array-based methods.

In our previous publication, we karyotyped and performed WGS of the parent line, analyzed with our SV pipelines, and microarray (aCGH) of the lines used (ref 31; *Tai et al.*, 2016, Nature Neuroscience). Given that there were no karyotypic changes compared to the parent line or aneuploidies of the mutation lines screened, any anomalies cryptic to these approaches are unlikely to influence the neuronal phenotypes detected across all lines studied.

We performed a new high resolution SNP array analysis to verify that the karyotypes of the iPSC-lines used in this study were unchanged compared to the previous study (ref 31). We did not detect any changes between the CRISPR-edited clones compared to the parental iPSC line GM08330 or compared to the previous study aCGH assay (Tai et al, 2016, Nature Neuroscience, ref 31, **Supplemental Table 1, Sheet 1-2**). Any micro changes detected with the SNP array are most likely features associated with the GM08330 background and shared across all lines. One shared detected mosaicism between control, 16pdup, and 16pdel lines is of 20q11.21CNV, a recurrent CNV found in iPSCs across several laboratories (*Martins-Taylor K.*, 2012, Stem Cells) and associated with increased Bcl-xL expression and increased survival advantage of pluripotent stem cells (*Nguyen*, et al., 2013, Molecular Human Reproduction). These changes are unlikely to affect to the neuronal phenotypes that we analyzed in this study since these changes were present across all the studied iPSC lines, including the naïve control line GM08330. Also, we did not detect any differences in the expression levels of *Bcl-xL (BCL2L1)* between different genotypes of the iPSC-derived DA neurons with RNAseq analyses (Control 6.8 ± 0.24 , 16pdup 6.6 ± 0.27 , 16pdel 6.7 ± 0.21 , mean expression \pm SD, **Supplemental Table 1 Sheet 3, and Supplemental Tables 2-3, Sheet3**). We added these SNP array results of the iPSCs lines in the **Supplemental Table 1, and page 5**.

3. (Gene expression analyses) Which genes from the 16p11 region are already expressed at the iPSC stage or the NPC stage in the present study?

Will differential expression affect pluripotency or early differentiation?

We analyzed expression of the genes in the 16p11.2 region with qRT-PCR from the iPSCs and NPCs, this data has now been added in the **Supplemental Figure 1 C-D, and page 5**.

The following genes are expressed at the undifferentiated iPSC stage (whose Ct values are <34): *TAOK, PPP4C, CDIPT, KCTD13, MAPK3, DOC2A, TMEM219, SEZ6L2, MAZ, PAGR1A, YPEL3, QPRT, PRRT2, ASPHD1, c16orf54, KIF22, SLX1A, ALDOA, TBX6, CORO1A, HIRIP3, FAM57B, MVP, INO80E, GDPD3, SPN*. We didn't detect any effects of the differential expression of these genes on the pluripotency of the iPSCs with different genotypes. Pluripotency of the iPSCs clones is shown in the **Supplemental Figure 1** with immunocytochemical staining of OCT4/TRA1-60 and NANOG/TRA1-60 and qRT-PCR of *OCT4* and *NANOG* (**Figure 2**).

The following genes are already expressed in the NPC stage (whose Ct values are <34): *TAOK, PPP4C, CDIPT, KCTD13, MAPK3, DOC2A, TMEM219, SEZ6L2, MAZ, PAGR1A, YPEL3, QPRT, PRRT2, ASPHD1, c16orf54, KIF22, SLX1A, ALDOA, TBX6, CORO1A, HIRIP3, FAM57B, MVP, INO80E, GDPD3, SPN, SULT, BOLA2A*. (**Supplemental Figure 1.D, page 5**).

During the early differentiation of the NPCs, we detected minor differences in the neuronal differentiation capacity of the NPCs at day 5 and day 18, shown in the **Figure 2** with increased TUJ1 expression in the 16pdel cells. In addition, we detected reduced FOXA2 expression in 16pdel and 16pdup populations at day 18 (**Figure 2**). We discuss these differentiation deficits in the manuscript (**pages 6-7, 17-19**), and these results are also consistent with previous study of the zebrafish and mouse models that display deficits in the neuronal development and DA neuron levels in 16pdel animals (*Pucilowska J, et al., 2015, J. Neurosci; Blaker-Lee A, et al., 2012, Dis. Model Mech.; Portmann, et al., 2014, Cell Rep.; Horev G., et al., 2011, PNAS*).

4. Did the authors perform pathway analysis based on their data from differential gene expression analysis (e.g. as in Ref. 14)? Did unbiased pathway analysis point to small GTPase (Rho) signaling (present study) or to MAPK/ERK signaling (shown in 16p11 CNV mouse models, e.g. Pucilowska, J Neurosci, 2015; Blizinsky, PNAS, 2016)?

To address this question, we performed a pathway analysis using ingenuity pathway analysis (IPA). Pathway analysis did not reveal any changes in Rho signaling, but ERK pathway was detected in the gene network of Module 19 among genes increased in 16pDup neurons and reduced in 16pDel neurons. IPA of top 5 disease categories in Module 25 revealed genes associated with 1) Cell-to cell signaling and interaction, nervous system development and function, 2) Cancer, dermatological diseases, organismal injury and abnormalities, 3) Neurological disease, 4) Psychological disorders, and 5) Behavior. These results of 16pdel DA neurons are consistent with the previous study of Kizner et al.(2020) that detected the same disease categories with RNAseq of *KCTD13*-Knock-out iPSC-derived neurons. (These new data have been added in the new **Supplemental Figure 4, Supplemental Tables 7-8, pages 8-9**).

Are any of the functional categories they identified by weighted co-expression analysis and gene ontology analysis (e.g. cell death, oxidative stress, metabolic processes etc.) affected in the 16p11 CNV iPSC-derived dopaminergic neurons?

We describe the results of the gene ontology analyses of the Modules 19 and 25 in the **Supplemental Tables 2-3**, and network pathway analyses in **Supplemental Tables 7-8, and page 8-9**.

To analyze metabolic processes in the DA neurons, we analyzed expression of metabolic marker LDHA in the sorted DA neurons at day 60 of differentiation. We detected increased LDHA expression in the 16pdel neurons compared to control or 16pdup neurons. LDHA levels were also slightly increased in 16pdup neurons compared to control cell population (data added in the **new Supplemental Figure 8**).

We did not detect any significant changes in mTOR-pathway activation as assessed by phosphoS6/total S6 ratio between different genotypes (data not shown). We did not analyze the cell death or oxidative stress in the DA-neurons in this study, but this is an interesting question for future studies.

5. The authors analyzed differential expression of selected genes from the two modules and discuss their potential (patho)physiological function in detail. Might any of these deregulated genes (e.g. GRIA4, KCND2, or others identified by RNA-seq) contribute to increased excitability in 16p11 del neurons and represent a druggable therapeutic target?

Are there any data regarding the potential side effects of long-term, systemic inhibition of RhoA signaling? Please discuss.

We now added discussion of these genes *GRIA4*, *KCND2* and their effects to excitability in the discussion section of the manuscript: **pages 19-20:**

“GRIA4 was also upregulated in 16pdel neurons. GRIA4 produces Glutamate-Ionotropic Receptor AMPA type Subunit 4, and it affects excitability of neurons by regulating synaptic development and plasticity^{66, 67}. AMPA receptor and GRIA4 have also associated with neurodevelopmental disorders with and without seizures⁶⁸ and schizophrenia⁶⁹, and drug and alcohol abuse⁷⁰. Thus, upregulation of GRIA4 may be involved in increased excitability of 16pdel neurons, and could be an interesting target for future drug screening studies for 16p11.2 CNV disorders.”

“Compared to 16pdel neurons, we detected decreased expression of these genes in 16pdup neurons, along with decreased expression of KCND2 and opioid receptor OPRM1. KCND2 contributes to neuronal plasticity and affects memory, learning and cognitive flexibility^{71, 72, 73} and it has been associated with development of ASD⁷⁴, Fragile X syndrome⁷⁵, and epilepsy⁷⁶. KCND2 regulates dendritic excitability via activation of synaptic NMDA receptors and controlling the expression levels of synaptic NR2B/NR2A fraction⁷⁷. This makes KCND2 interesting target molecule for future pharmacological studies of 16p11.2 CNV phenotypes.”

“No adverse effects were detected in stressed or non-stressed animals when treated with Rhosin for several days⁹¹. Rhosin enhances neurite outgrowth and also reduces cell growth and endothelial cell invasion via blocking RhoA activity, without affecting Cdc42 or Rac1 signaling⁹². This pathway specificity raises the possibility of Rhosin as a potential candidate for treatment of neuropsychiatric and neurodevelopmental disorders. However, to date no long term safety studies with Rhosin have been conducted in animal models with neurological deficits.” (This discussion has been added in manuscript **page 23**).

6. In Figure 3 A, C numerous Hoechst-positive nuclei are visible that do not show TUJ1/TH immunoreactivity. The authors should indicate the percentage of non-neuronal (TUJ1 negative) cells after NCAM sorting and maturation.

Are there any hints from their RNA-seq analysis, which other cell types are detectable in these cultures? How did the authors ascertain that they record from dopaminergic neurons during patch-clamping (e.g. by transduction with a TH_GFP lentivirus)?

-To answer these questions, we quantified the percentage of non-neuronal (TUJ1-negative) cells from the population. The percentage of TUJ1-negative cells was 13-15% from total cell population in different genotypes (14.2±10 % in ctr, 15.2±11 % in 16pdup, 13.7±6.1 in 16pdel, average ±SD) , and this data has been added in the **Supplemental Figure 3, and page 7**.

-We analyzed our RNAseq data to identify which other types of cells are present in the cell population; we detected genes specific for human midline progenitors, floor plate progenitor cells, dopaminergic

neuroblasts, mediolateral neuroblasts type 1, and oculomotor and trochlear nucleus. This data is consistent with previous single cell analyses of hESC-derived DA neuron and iPSC-derived DA neuron cultures (La Manno, et al 2016, Cell).

-Based on the gene expression analyses and immunocytochemical characterization we detected 13-26% SOX2/Nestin positive neural progenitor cells in the cell population at day 60 (**Supplemental Figure 3, page 7**), and no difference between different genotypes were detected.

-We also detected small percentage of GABA-positive cells in all the neuronal cultures at day 60 (14-21% from total cell population, and no difference between different genotypes were detected (**Supplemental Figure 3, page 7**).

-We did not detect any CTIP2-positive cortical neurons, nor H9B-positive motoneurons in the cultures at day 60 (**Supplemental Figure 3, page 7**).

-We did not detect any GFAP-positive astrocytes in the cell population at day 60 (**Supplemental Figure 3, page 7**).

-The patch clamp recordings were done with a patch-pipette loaded with Alexa-488 dye, and the dye passively diffused into cells during the patching. To verify the identity of the patched neurons, we monitored the neuronal morphology of the DA-neurons, and after patching fixed the cells and stained them with tyrosine hydroxylase (TH). We did not use neurons that were transduced with TH-GFP-virus for these patch clamp experiments.

7. Figure 7 shows a clear trend towards increased RHOA protein expression in 16p11 dup iPSC-derived neurons compared to controls, however, this trend is not visible in the firing rate. Please discuss.

Although we detected a trend towards increased RHOA protein expression in the 16p11.2 dup neurons, this increase was not statistically significant compared to control neurons (**Figure 7C and 8B**). Thus, the small change in the RHOA expression was not sufficient to induce changes in the firing rate of the 16pdup neurons compared to control neurons (**Figure 3-7**).

We added this discussion in the results section **page 14**: *“Although we detected a trend towards increased RHOA expression in the 16pdup neurons compared to controls, this change was not statistically significant (Figure 7C, 8B).”*

The authors suggest that the decrease in KCTD13, which acts as an adaptor protein for Cullin-3 ubiquitin ligase, underlies the increase in the substrate RHOA. To make this more convincing, protein expression of Cullin-3 should be analyzed by immunoblotting as well.

We now added the western blot analyses of Cullin-3 expression in the new **Figure 8., page 14**. *“We also measured the Cullin3 expression in these cell populations and detected increased expression of it in the 16del neurons compared to control cells (Figure 8A-B), which indicates that reduction of KCTD13 plays key role in the KCTD13-Cullin3 complex that facilitates the increase of RHOA activation in the 16pdel neurons.”*

8. (Discussion) Portmann et al (Ref. 57) and Horev et al (PNAS, 2011) analyzed alterations in dopamine-modulated brain circuits and midbrain dopaminergic neurons in 16p11 CNV mouse models. Their findings are highly relevant to the present study and should be discussed.

We appreciate this helpful comment. We now refer to these findings in the discussion of the manuscript **page 17**:

“These studies have shown that mice with heterozygous 16pdel had significantly increased midbrain region⁵¹ and deficits in basal ganglia- and DA-regulated circuitry, including increased numbers of Drd2⁺ medium spiny neurons in the striatum and downregulation of DA signaling components (Darpp32 and Drd1) in deeper cortical layers⁴⁸. The Drd2⁺ MSNs also displayed synaptic deficits with increased EPSCs, and 16pdel mice also had reduced Th expression in the mesodiencephalon, which implicated deficits in dopamine signaling system⁴⁸. In addition, 16pdel and 16pdup mice displayed opposite behavioral phenotypes, and these behavioral deficits were more severe in 16pdel mice⁵¹. 16pdel mice had poor adaptation to change, hyperactivity, learning deficits, repetitive and restricted behavior, sleep problems, deficits in movement control and hearing^{48, 51}.”

9. (Discussion) It should be mentioned that RhoA levels were unchanged in the cortex and hippocampus of 16p11 del mice and decreased in 16p11 dup mice in the study by Arbogast et al (Ref. 77).

We now added this reference in the discussion **page 22**.

“In addition, the RhoA levels were unchanged in the hippocampus and cortex of the 16pdel mice compared to control mice, whereas RhoA levels were decreased in the 16pdup mice (Argogast et al., 2019).”

10. Please provide a web link for your publicly available RNA-seq dataset.

Thank you for this suggestion. We will add a web link to our RNA-seq dataset once the manuscript is in press.

Minor points

11. The authors performed NCAM sorting to enrich iPSC-derived neurons. Although cell sorting will decrease heterogeneity, it may also obscure ASD-relevant, early neurodevelopmental alterations induced by 16p11 deletion/duplication. This downside should be discussed.

We added the following paragraph in the discussion section of the manuscript, **page 18**.

“We differentiated and enriched DA neurons from iPSCs with NCAM^{high+} sorting that resulted in homogeneous DA neuron populations of the control, 16pdel, and 16pdup cells. The limitation of generating homogeneous neuronal cell populations is that we may have dismissed some developmental cellular deficits that may be relevant for ASD-phenotypes. Such deficits could include proliferation of precursor cells, differential generation of various neuronal subclasses e.g. number of excitatory vs inhibitory neurons^{52, 53}, and alterations in astroglial cell development that is present in some ASD-phenotypes and occurs in unsorted iPSC derived neural cell populations during long term culturing *in vitro*⁵⁴. However, production of enriched DA neuron populations enabled us to detect disease phenotypes in a specific neuronal population that has a key role in the development of the 16p11.2 CNV disorders.”

12. In Figure 1H large aggregates of TH/TUJ1-positive iPSC-neurons are visible, which are notoriously difficult to quantify by HCI analysis. The authors should show a representative segmentation of this microscopic image by their HCI analysis software.

Thank you for pointing this out. To improve the presentation of the quantified neuronal cell populations we added additional batch of staining and quantification of TH/TUJ1 in the **Figure 1 I- J** and added IXM images of quantified cell areas with less cell aggregates. In addition, we added representative cell masks

of the IXM algorithm that were used for quantification of the TH/TUJ1-positive neurons in these cultures (new Supplemental Figure 6).

13. Spike rate and burst rate of wild-type iPSC-derived neurons on MEA usually increase during maturation. The authors should discuss, why this increase is not detected in the present study.

When we look at the early development of the network activity between days 0-22 after plating to LD-MEA, we do see an increase in the spike rate and burst rate of the wild-type iPSC-derived networks (Figure 7). Spike rate in the wild-type neurons was 259 ± 51 spikes (day 4) and it increased to 704 ± 112 spikes (day 13), and up to 1022 ± 124 spikes (day 15) to 910 ± 196 spikes (day 22), after which the spike rate didn't increase further, and it decreased to 401 ± 99 spikes (day 28). Number of bursts in the wild-type neurons increased to 4.3 ± 1.8 (day 17) and 4.5 ± 2.5 (day 24) after which it decreased to 3.8 ± 2.8 bursts (at day 28) (Figure 7).

We added discussion about this network development in manuscript page 15-16:

“Although we detected increase in the number of spikes over time in the control DA neuron networks at day 4 after plating (259 ± 51 spikes) until day 15 (up to 1022 ± 124 spikes), the spike rate plateaued between days 15-22, and then was reduced to 401 ± 99 spikes at day 28. After culturing the neurons over three weeks on MEA, the DA neuron networks may be getting more sparse due to neuronal migration and accumulation into clusters, which may affect the overall activity of the neurons after day 22.”

14. It might be helpful for many readers to provide a supplementary table listing disease-relevant findings in iPSC models / mouse models of the 16p11 del syndrome and in human 16p11 del carriers diagnosed with ASD (SCZ).

Thank you for this suggestion. We added **Supplementary Table 9** in the manuscript to describe the disease phenotypes of patients with 16p11.2 CNV, mouse models of the 16pdel syndrome and iPSC-derived neurons with 16p11.2 CNVs.

Reviewer #2 (Remarks to the Author):

The manuscript by Sundberg et al. describes the thorough characterisation of isogenic iPSC lines with deletions and duplications in the 16p11.2 locus, which is associated with the development of schizophrenia and autism. The study is well-performed, containing a thorough characterisation of the neurons through RNA sequencing, analysis of synapse formation, electrophysiology and MEA. The study shows altered electrophysiological features, and in particular altered burst firing patterns in the 16p11.2 CNV neurons. Many of the observations are in line with observations from a previous study (Deshpande et al., 2017), also studying 16pDel and 16pDup in human iPSCs. Nonetheless, the observations also provide some unique and relevant insights to the potential pathophysiology of neuropsychiatric disorders through RNA sequencing and MEA measurements. The study further shows that some of the observed abnormalities can be rescued by treatment with the Rho inhibitor Rhosin. The latter is however not novel, as it builds on a previous finding by Escamilla et al, published in Nature in 2017. Overall, the study contributes with solid and thorough validation in a human isogenic model system of previously presented hypotheses in the field, and it provides some new insights regarding RNA expression patterns, synchronous activity and burst firing patterns in the mutated cell lines. A few points require further attention:

The study emphasises the relevance of studying the 16p11.2 CNVs in the context of midbrain DA

neurons, which are known to play a key role in neuropsychiatric disorders. This is further a key feature of the current study to distinguish from the Deshpande et al., study which uses iPSCs. However, although the authors show that the progenitor cells in the cultures express FOXA2 at an early stage (day 5. Fig. 1C+E), there is not provided much evidence to support that the DA neurons in the cultures are truly of a midbrain DA phenotype. The FOXA2 expression in the cultures seems for some reason to be unstable, since a significant proportion of the cells are FOXA2-neg at 17 days of differentiation. **Please provide stainings of FOXA2/LMX1A and of NURR1/TH cells at a late time point in the culture (i.e. day 60) + and quantifications of at least one of these stainings.**

Thank you for this important question. We performed FOXA2/LMX1A and NURR1/TH immunostaining, and we quantified the number of NURR1/TH expressing cells at day 60. We detected ~63% of NURR1/TH double positive cells in the cell populations, and no significant differences between genotypes were detected. This data has been added in the new **Supplemental Figure 3, page 7**. “To verify the ventral midbrain identity of the DA neurons, we quantified TH/NURR1 expression at day 60: 62.9±10 % control, 62.9±9.4 % 16dup, and 63.8 ±14.6 % in 16pdel, data are presented as average±SD. We also detected co-expression of LMX1A/FOXA2 in these cell populations (**Supplemental Figure 3A-C**).”

We also verified the expression of DA neuron specific genes *TH* (Figure 2A), *LMX1B*, *GIRK2*, *AADC*, *NURR1* (Supplemental Figure 3) in the cell cultures with RNAseq (**page 8**).

It would be beneficial to show the differentiation protocol with addition of the different factors in the schematic in Fig., 1A

We now added the differentiation protocol and factors in the schematic **Figure 1A**.

From the methods section, it appears that “Three clones from each genotype were used for the DA neuron differentiation”. It also appears that “ All experiments were repeated at least three times with independent differentiation batches of neurons”. It is however unclear if all three clones were used for all experiments, and how many biological replicates are included for each clone in each experiment.

For some of the key experiments, the data should be shown independently for each clone in order to validate that all the clones from the same genotype are displaying the same phenotype. This is to ensure that it is not one or two clones in the dataset which are driving the differences.

Thank you for raising this important question. We now added in the **materials and methods** section detailed information of the number of cell lines, genotypes and number of clones used in each experiment. In addition, we added individual clone data of three clones per genotype for the HD-MEA experiments, (new **Supplemental Figure 7**).

We also added individual clone data of three clones per genotype for the TH-positive DA neuron soma size experiments (**new Supplemental Figure 8**), and we included representative blots of western blot analyses with vehicle/rhoin treatments from three individual clones per genotype (**Supplemental Figure 8**).

For each analysis it should be noted in the figure legend or materials and methods how many clones were used and how many biological replicates for each clone are included in the data.

Thank you for this comment. We included the information of the number of clones used and number of biological replicates used in the **materials and methods** section.

The manuscript is very lengthy and could benefit from significant trimming. In particular, there are several references to each individual observation in the results and discussion sections, which leaves an impression of a scattered manuscript with lack of focus.

We condensed the text in several sections, in particular results section pages 8-9, figure legends, and discussion section page 19.

Line 143: "of these neurons >50-75% were FOXA2 positive". Should read either ">50%" or "50-75%"
Thank you for pointing this out. It is now corrected in the text: 50-75%.

Reviewer #3 (Remarks to the Author):

In the present manuscript, Sundberg M et al utilize dopaminergic neurons derived from human iPSCs carrying CNVs of 16p11.2 duplication and deletion induced by CRISPR-Cas9 gene editing to investigate network functions and alterations seen in neuropsychiatric/neurodevelopmental disorders as schizophrenia and autism. In this work, neuronal differentiation is well characterized, and cultures are enriched for neurons with a FACS step previous gene expression analysis, electrophysiological recordings and network investigations. Electrophysiological assessments are performed both with whole cell patch clamp recordings and, at network level, with MEA chips. The authors report that 16p11.2 deletion results in synaptic connectivity and excitability changes in dopaminergic neurons which lead to hyperactive networks in vitro. RhoA inhibition obtained with Rhosin administration rescued network hyperactivity.

Overall the work is sound, well designed, and of interest for the field.

In my opinion, few points should be addressed before considering this manuscript for publication:

-The Results section is lacking details about the Statistics used to analyze different experiments as well as N numbers. This information should be included.

-We thank the reviewer for these comments. To keep the length of manuscript as compact as possible, we included the information about the statistical methods used **in the figure legends** of each experiment (in few cases, this information is also included in the results section). We added the number of N per experiment in **the materials and methods** section under each experimental method.

-In Figure 1E-G, the authors report an increased number of Tuj1 positive cells in the deletion condition, compared to control and duplication. Same for the NCAM analysis in G. However, they report that the ratio between Tuj1 and TH positive neurons is maintained among the different conditions. Do the authors know the fate of these "other" neurons and to which extent are they present after FACS enrichment? The data seem to suggest that these neurons are not dopaminergic. This could potentially play a role in the network studies. This point should be clarified.

-We thank the reviewer for raising this important question. To address this question, we performed extensive analyses of the neuronal populations after FACS sorting and differentiation.

-First, we analyzed cell type specific gene expressions of the iPSC-derived DA neuron cultures at day 56 of differentiation to identify which other types of cells are present in the cell population. We compared our RNAseq data to previously published human fetal and mouse midbrain RNAseq data (*La Manno, et al 2016, Cell*). We detected genes specific for human midline progenitors, floor plate progenitor cells, dopaminergic neuroblasts, mediolateral neuroblasts type 1, and oculomotor and trochlear nucleus, and DA neurons. This data is also consistent with previous single cell analyses of hESC-derived DA neuron and hiPSC-derived DA neuron cultures (*La Manno, et al 2016, Cell*).

-With immunocytochemical staining and imaging, we detected ~85-88 % of TUJ1-positive neurons in these cell populations, and 13-15% TUJ1-negative cells, and from these populations we identified SOX2/Nestin-positive neural precursor cells: 13-26 % from total cell populations at day 60 (**Supplemental Fig3**). From the TUJ1-negative cells we did not detect any GFAP positive astrocytes (**Supplemental Fig3**). From the TUJ1-positive neurons we did not detect any CTIP2-positive cortical neurons, or H9B-positive motoneurons (**Supplemental Fig3**). We identified TUJ1-positive neurons that were GABA-positive neurons; these ranged ~14-21% from total cell populations in different genotypes (**Supplemental Fig3**). Since we did not detect any statistically significant differences in the GABA-positive cell populations between different genotypes, we think that these cells didn't play significant role in the disease phenotypes of the network activity between different genotypes. We added this new data in the new **Supplemental Figure 3** and **page 7** in the manuscript.

- "To verify the ventral midbrain identity of the DA neurons, we quantified TH/NURR1 expression at day 60: 62.9±10 % control, 62.9±9.4 % 16pdup, and 63.8 ±14.6 % in 16pdel, data are presented as average±SD. We also detected co-expression of LMX1A/FOXA2 in these cell populations (Supplemental Figure 3A-C). Number of TUJ1-negative cells were: ~13-15% from total cell population. We identified that these cells were SOX2/Nestin-positive progenitors: 18.3±4.6% control, 13.1±4.1 % 16pdup and 26.3±9.4 % in 16pdel (data presented as average ±SD, Supplemental Fig 3D). We also detected GABA positive neurons among the neuronal cell populations: 20.7±11.5 % in control, 18.8±5.6% in 16pdup, and 13.7±5 % in 16pdel. No differences between different genotypes were detected in the expression of these markers (Supplemental Figure 3D). We did not detect any CTIP2+ cortical neurons, or H9B+ motoneurons, or GFAP+ astrocytes at day 60 of differentiation (Supplemental Figure 3J-L)."

-Heatmaps in Figure C and D are not of clear reading and differences in gene expression are not always obvious. I suggest to use a lighter color for the 0 value instead of black.

-To clarify the presentation of the data in the **Figure 2**, we revised the heatmaps of the transcriptional expression data. We adjusted the colors on these heatmaps to red, blue and lighter colors, and we also used lighter color for the 0 value (updated **Figure 2**).

-Figure 3C, differences in SYP1 densities in the three conditions are difficult to appreciate, I suggest using different microphotographs.

-We thank the reviewer for this comment. To clarify the presentation of the SYP1-puncta in the immunocytochemical pictures of the SYP1/TH stained neurons, we added more representative confocal microscope images in the **Figure 3C**.

-Figure 4, since the authors report differences in soma area in the 16p11.2 deletion condition, did they see also differences in cell capacity from whole cell recordings? Also, a separate bar graph should be add with Rheobase quantifications.

-Thank you for raising this important point of clarification. Cell capacitance measurements were performed during the patch clamp experiments. Although slightly smaller soma size were detected visually with the Alexa dye in the control cell population (Figure 4B), this difference was not significant when comparing cell capacitance. We added the capacitance numbers in the text, **pages 11-12**.

-Additionally, per the reviewer's suggestion, we now include Rheobase values in bar graph format in **Figure 4E**.

-Figure 7, the functional data obtained after Rhosin administration should be strengthen with investigations on the morphological and molecular changes of the dopaminergic neurons within the network. Moreover, RhoA inhibition should be validated in the culture to exclude off target effects.

-We treated the DA-neuron cultures after cell sorting with Rhosin or DMSO (vehicle) for 23 days during

differentiation, and evaluated their morphology at day 60 of total differentiation time (new **Figure 8**). We calculated the soma size of TH-positive DA neurons and detected significantly increased soma area in the 16pdel neurons treated with vehicle compared to other groups. Rhosin treatment decreased the soma size significantly in the 16pdel DA-neurons (this new data has been added in the new **Figure 8, page 16**). We also analyzed number of primary neurites per TH-positive DA neurons, but we did not detect any significant changes between cell groups or veh/Rhosin treatments (data not shown).

- To validate RhoA inhibition in the DA neuron cultures after Rhosin or vehicle treatment, we analyzed the total RHOA expression levels with western blot method. RHOA expression was increased in the 16pdel neurons treated with vehicle compared to other conditions. Rhosin treatment reduced RHOA levels in the 16pdel neurons (this new data has been added in the **new Figure 8, page 16**).
- We also measured the phosphorylation of Cofilin which is a downstream target for RHOA-pathway activation (RHOA-LIMK2-cofilin), and we detected increased expression of phospho-cofilin in the 16pdel neurons treated with vehicle compared to other groups. Rhosin treatment reduced the phospho-cofilin in 16pdel neurons, validating its RHOA-pathway specific effects in neurons, this data has also been added in the **new Figure 8, page 16**.

Reviewers' Comments:

Reviewer #1:

Remarks to the Author:

Reviewer #1

The revised manuscript has clearly improved with additional information and data fulfilling the points raised in the first round of reviews. Therefore, I would recommend to publish this interesting manuscript in Nature Communications.

Reviewer #2:

Remarks to the Author:

The authors have adequately addressed reviewer comments in their revised version of the manuscript, and I have no further concerns. I would only ask that the new data presented in Fig S8C (soma size of neurons shown in separate clones) is compiled into one single graph where the data from the three clones of each genotype is shown side-by-side, since the main purpose of showing clonal data is to be able to compare each clone directly to each other as well as to the clones of the mutated genotypes.

Reviewer #3:

Remarks to the Author:

The authors have added extensive analysis and the manuscript is significantly improved. All my initial concerns have been addressed and in my opinion this work is now suitable for publication.